# ClinBench: A Standardized Multi-Domain Framework for Evaluating Large Language Models in Clinical Information Extraction

Ismael Villanueva-Miranda[✉,1], Zifan Gu[1], Donghan M. Yang[1], Kuroush Nezafati[1], Jingwei Huang[1], Peifeng Ruan[1], Xiaowei Zhan[1], Guanghua Xiao[1,2,3], and Yang Xie[✉,1,2,3]

[1]Quantitative Biomedical Research Center, Department of Health Data Science and Biostatistics, Peter O'Donnell Jr. School of Public Health, The University of Texas Southwestern Medical Center, Dallas, TX 75390, United States
[2]Department of Bioinformatics, UT Southwestern Medical Center, Dallas, TX 75390, USA
[3]Simmons Comprehensive Cancer Center, UT Southwestern Medical Center, Dallas, TX 75390, USA
✉Corresponding authors: {Yang.Xie,Ismael.Villanueva-Miranda}@UTSouthwestern.edu

## Abstract

Large Language Models (LLMs) offer substantial promise for clinical natural language processing (NLP); however, a lack of standardized benchmarking methodologies limits their objective evaluation and practical translation. To address this gap, we introduce **ClinBench**, an open-source, multi-model, multi-domain benchmarking framework. ClinBench is designed for the rigorous evaluation of LLMs on important structured information extraction tasks (e.g., tumor staging, histologic diagnoses, atrial fibrillation, and social determinants of health) from unstructured clinical notes. The framework standardizes the evaluation pipeline by: (i) operating on consistently structured input datasets; (ii) employing dynamic, YAML-based prompting for uniform task definition; and (iii) enforcing output validation via JSON schemas, supporting robust comparison across diverse LLM architectures. We demonstrate ClinBench through a large-scale study of 11 prominent LLMs (e.g., GPT-4o series, LLaMA3 variants, Mixtral) across three clinical domains using configurations of public datasets (TCGA for lung cancer, MIMIC-IV-ECG for atrial fibrillation, and MIMIC notes for SDOH). Our results reveal significant performance-efficiency trade-offs. For example, when averaged across the four benchmarked clinical extraction tasks, GPT-3.5-turbo achieved a mean F1 score of 0.83 with a mean runtime of 16.8 minutes. In comparison, LLaMA3.1-70b obtained a similar mean F1 of 0.82 but required a substantially longer mean runtime of 42.7 minutes. GPT-4o-mini also presented a favorable balance with a mean F1 of 0.81 and a mean runtime of 13.4 minutes. ClinBench provides a unified, extensible framework and empirical insights for reproducible, fair LLM benchmarking in clinical NLP. By enabling transparent and standardized evaluation, this work advances data-centric AI research, informs model selection based on performance, cost, and clinical priorities, and supports the effective integration of LLMs into healthcare. The framework and evaluation code are publicly available at https://github.com/ismaelvillanuevamiranda/ClinBench/.

## 1 Introduction

Large Language Models (LLMs) are becoming increasingly important in clinical natural language processing (NLP), often outperforming traditional methods on tasks such as clinical information

extraction and rare medical event detection [1–3]. However, the rapid adoption of LLMs in clinical applications has outpaced the development of systematic evaluation methods. Current evaluations typically rely on specialized datasets or informal testing protocols, lacking a unified, structured approach to evaluating the practical utility and generalization of LLMs across various clinical domains [4, 5]. Although initial benchmarking exists for broader biomedical NLP tasks [6, 7], these efforts do not adequately address the specific challenges of clinical information extraction, such as handling diverse medical terminology across multiple clinical areas.

Clinical reports from areas such as oncology, electrocardiography (ECG), and Social Determinants of Health (SDOH) significantly differ in their terminology, writing style, and data formats [8, 9]. Additionally, privacy restrictions associated with proprietary cloud-based LLMs [10–12], and resource or performance issues in open-source models, present major challenges to reliably selecting and deploying these models in clinical settings. Thus, there is an urgent need for a clear and comprehensive benchmarking framework specifically designed to evaluate LLMs in clinical information extraction tasks.

To bridge these gaps and promote Data-Centric AI Research in clinical NLP, we introduce **ClinBench**, an open-source benchmarking framework designed to assess multiple LLMs across different clinical domains systematically. ClinBench evaluates the abilities of the models to extract structured clinical information from medical texts through standardized methods. The framework defines key evaluation components, including data requirements, prompt configurations specified in YAML, JSON-based output validation, and a set of detailed performance metrics (e.g., F1 score, runtime).

To demonstrate the effectiveness of ClinBench, we conducted an extensive benchmarking study involving 11 widely used LLMs, both open source and proprietary. This study evaluated models using complex clinical information extraction tasks from three publicly available clinical datasets: (1) lung cancer staging from TCGA pathology reports, (2) atrial fibrillation detection from MIMIC-IV-ECG interpretations, and (3) extraction of SDOH factors from MIMIC clinical notes.

The main contributions of this paper are:

- **ClinBench Framework:** We introduce a modular, open-source framework designed for standardized, reproducible evaluations of LLMs across diverse clinical information extraction tasks.
- **Structured Multi-Domain Benchmark Tasks:** We provide a reproducible and extensible set of standardized tasks across three distinct clinical domains using publicly available datasets (TCGA, MIMIC-IV-ECG, MIMIC clinical notes).
- **Extensive Quantitative Analysis:** We systematically compare 11 LLMs, highlighting their performance metrics (such as F1 scores) and practical efficiency (runtime and cost) across diverse clinical contexts.
- **Openly Available Resources:** All components of ClinBench, including evaluation methods, prompt templates, benchmark results, dataset details, and metadata, are publicly accessible to encourage further research and transparency.

## 2   Related work

Large Language Models (LLMs) are increasingly applied to clinical information extraction, with studies demonstrating their utility for tasks ranging from identifying rare disease phenotypes [13] to few-shot extraction from general clinical notes [14]. A significant body of research also focuses on enhancing LLM performance through advanced prompt engineering, in-context learning strategies, or instruction tuning specifically for information extraction [15–18]. While these contributions highlight LLM capabilities and refined interaction methods, evaluations are often designed for specific tasks or models and typically do not provide a broader, standardized framework for comparing diverse LLMs across multiple clinical domains, especially with respect to practical aspects such as computational efficiency and the reproducibility of the benchmarking process itself.

Efforts to systematically benchmark LLMs for medical applications are emerging. For instance, MedGPTEval provides an evaluation system and datasets for LLMs in the Chinese medical context [19], while LLM-AIx offers an open-source pipeline for information extraction using locally deployable LLMs, demonstrated on tasks such as anonymization and TNM staging [20]. Other research has introduced benchmarks for fine-grained information extraction, emphasizing the importance of

detailed instructions and evaluating generalization [21]. These initiatives are valuable for assessing LLMs in specific contexts or with particular deployment considerations.

However, there remains a clear need for an open-source, easily extensible framework specifically designed to systematically evaluate a wide array of state-of-the-art LLMs, spanning both proprietary and open-source architectures, across multiple clinical domains, using standardized configurations of publicly available English-language datasets. Such a framework should also incorporate computational efficiency as a key evaluation metric and ensure reproducible output structures through mechanisms such as schema validation.

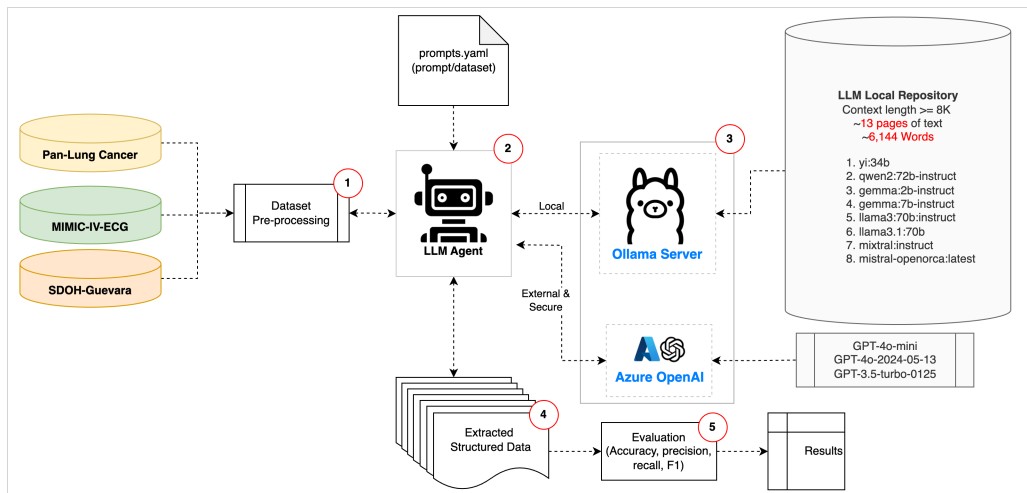

Figure 1: The **ClinBench** automated five-step benchmarking workflow. This diagram illustrates the standardized pipeline from (1) ingestion of preprocessed clinical datasets, through (2) YAML-configured, schema-guided LLM orchestrator processing and (3) model inference (local/API), to (4) JSON-validated data extraction, and (5) comprehensive performance evaluation using standardized metrics.

## 3 The ClinBench framework

This paper introduces **ClinBench**, an open-source, multi-model, multi-domain benchmarking framework designed for the systematic, reproducible, and standardized evaluation of Large Language Models (LLMs) on clinical information extraction tasks. ClinBench implements a core five-step automated pipeline, as shown in Figure 1, that manages the evaluation process from data ingestion to performance reporting. This structured workflow ensures that each LLM is assessed under uniform conditions, promoting consistency and facilitating fair comparisons across diverse models and tasks.

The ClinBench workflow (Figure 1) begins by (1) ingesting task-specific datasets, which must conform to a predefined, standardized format (typically structured CSV files) to ensure a consistent evaluation baseline. (2) A central LLM Orchestrator agent then processes these datasets, dynamically loading standardized prompts from version-controlled YAML configuration files. These YAMLs are a key feature of ClinBench, encapsulating task instructions, relevant domain knowledge, and precise JSON schemas for the expected LLM output format. (3) The orchestrator subsequently manages model inference through a unified interface. This interface supports interactions with both local open-source LLMs (e.g., via Ollama, enabling complete data control) and proprietary API-based models via secure, institutionally compliant connections (e.g., via Azure OpenAI Service). This dual capability is handled by the same orchestrator, ensuring standardized context window utilization and uniform API call formatting to promote consistent evaluation conditions across model hosting environments. (4) LLM-generated outputs, produced in the JSON format specified by the prompt configurations, then undergo an automated validation step. This validation, facilitated by the strictjson library [22], rigorously checks each output against its corresponding predefined JSON schema to confirm structural correctness, data type adherence, and the use of permissible values, thereby ensuring data quality and output standardization before evaluation. (5) Finally, a dedicated evaluation module compares these validated outputs against expert-annotated ground truth, computing

a comprehensive suite of performance metrics. This entire pipeline is designed for robust, end-to-end execution.

Key design principles of ClinBench further enhance its utility and adaptability. The framework's unified orchestrator architecture for managing both local and secure remote model inferences (as detailed in Step 3) provides crucial flexibility for diverse research and deployment scenarios, including those with stringent data privacy requirements. Furthermore, ClinBench is built for extensibility; its modularity, particularly the reliance on externalized YAML prompt configurations and standardized data input formats, allows new datasets and LLM architectures to be readily integrated. This requires minimal modifications to the core evaluation pipeline, positioning ClinBench as a durable and evolving tool for ongoing research in clinical NLP.

# 4 Benchmark datasets

To demonstrate the ClinBench framework and establish a multi-domain benchmark, this study employs three distinct clinical tasks requiring structured information extraction. These tasks utilize curated configurations of publicly available datasets, ensuring the reproducibility and accessibility of our benchmark. Each dataset configuration, detailed below, presents unique challenges in text structure, clinical terminology, and extraction complexity, thereby contributing to a comprehensive evaluation of LLM capabilities.

## 4.1 Lung cancer staging from pathology reports

**Motivation and task:** Accurate extraction of tumor characteristics (pT, pN classifications) and overall stage from pathology reports is fundamental for oncological decision-making [23]. This task evaluates LLMs on extracting these cancer staging elements according to established guidelines.

**Data source and cohort configuration:** The data for this task are derived from Pan-Lung Cancer (TCGA, Nat Genet 2016) [24]. Information and aggregated data can be explored via portals such as the cBioPortal for Cancer Genomics (e.g., the NSCLC TCGA Broad 2016 study at `https://www.cbioportal.org/study/clinicalData?id=nsclc_tcga_broad_2016`). For ClinBench, the input comprises a specific cohort of 774 free-text pathology reports selected from this project. The list of unique identifiers for these 774 reports (e.g., case or file IDs) is provided as 'Lung_notes_id.csv' in our publicly available ClinBench code repository under the 'benchmark_data_definitions/tcga_lung/' directory. This allows users with appropriate access to the original TCGA data to extract the precise cohort used in our benchmark. The framework ingests these report texts. The corresponding annotation reference file, containing the expert-annotated ground truth for tumor characteristics (pT, pN, overall stage, and histologic diagnosis), is based on prior work [1] and aligned with AJCC 7th edition criteria [25].

**Extraction target and ground Truth:** ClinBench tasks LLMs with extracting four variables from the pathology report texts: primary tumor classification (pT), regional lymph node involvement (pN), overall tumor stage, and histologic diagnosis. The expert-annotated ground truth for these variables is provided during the evaluation step.

**Benchmark relevance:** This dataset configuration presents LLMs with highly specialized oncological language. It involves extracting various interrelated text components based on complex clinical classification systems, thereby testing a detailed understanding.

## 4.2 Atrial fibrillation detection from ECG reports

**Motivation and task:** Rapid, accurate identification of atrial fibrillation (AF) from electrocardiogram (ECG) interpretations supports timely clinical intervention [26]. This task evaluates an LLM's ability to extract AF presence from narrative ECG reports.

**Data source and cohort configuration:** This task utilizes electrocardiogram (ECG) interpretations from the MIMIC-IV-ECG Database v1.0 [27], available on PhysioNet at `https://physionet.org/content/mimic-iv-ecg/1.0/`. Access to MIMIC-IV-ECG requires PhysioNet credentialing, which involves completing human subjects research training and signing a data use agreement. For ClinBench, our collaborators selected a specific cohort of 700 ECG report texts from MIMIC-IV-ECG v1.0. This selection was performed to include only definitive atrial fibrillation (AF) or non-AF cases;

paced rhythm and uncertain interpretations were excluded to ensure unambiguous ground truths for evaluation, as per the original dataset documentation [27]. The text from these selected reports was used as input for our benchmark. The list of identifiers (e.g., 'note_id's) for these 700 reports is provided as 'ECG_notes_id.csv' in our publicly available ClinBench code repository within the 'benchmark_data_definitions/mimic_ecg_af/' directory. This allows users with appropriate access to MIMIC-IV-ECG v1.0 to extract the precise cohort used in our benchmark. The ground-truth labels for this binary classification (AF/NotAF) were established through manual domain-expert annotation, as described in the original MIMIC-IV-ECG dataset documentation [27].

**Extraction target and ground truth:** ClinBench tasks LLMs with classifying each ECG report as indicating "AF" or "NotAF". Ground-truth labels for this binary classification were established through manual domain-expert annotation (per the original dataset documentation [27]) and are aligned with the input texts.

**Benchmark relevance:** This dataset configuration tests LLMs on concise, semi-structured clinical reports with specialized terminology, focusing on a binary classification extraction important for cardiovascular assessment.

### 4.3   Social determinants of health (SDOH) factor extraction

**Motivation and task:** Identifying SDOH from unstructured clinical notes is important for understanding patient context and addressing health disparities [28]. This task evaluates an LLM's capability to extract key SDOH factors related to employment and housing.

**Data source and cohort configuration:** This task utilizes narrative clinical notes (discharge summaries) from the MIMIC-III Clinical Database (e.g., v1.4), a large, de-identified public-access database hosted on PhysioNet (`https://physionet.org/content/mimiciii/`). Access to MIMIC-III requires PhysioNet credentialing, which involves completing human-subjects research training and signing a data-use agreement. The ground truth annotations are derived from the MIMIC-SBDH dataset by Ahsan et al. (2021) [29]. This resource, available at `https://github.com/hibaahsan/MIMIC-SBDH`, provides annotations for 7,025 discharge summary notes from MIMIC-III. For ClinBench, we utilized a cohort of 1,405 unique discharge summaries selected from this MIMIC-SBDH dataset. The list of 'SUBJECT_ID' and 'HADM_ID' pairs identifying these specific admissions is provided as 'sdoh_subjects_id.csv' in our publicly available ClinBench code repository within the 'benchmark_data_definitions/mimic _sdoh/' directory. This allows users with appropriate access to MIMIC-III and the MIMIC-SBDH annotation files to recreate the cohort and ground truth used in our benchmark. The text from these selected discharge summaries was used as input. We did not perform additional preprocessing on the note texts beyond their original provision.

**Extraction target and ground truth:** ClinBench aims to extract two SDOH variables: (1) employment status (categorized as "Employed", "Unemployed", or "Unknown") and (2) housing conditions (classified as "Housing", "Homeless", or "Unknown"). Ground truth for these categories, derived from the original SDOH dataset annotations, is structured for direct comparison with LLM outputs.

**Benchmark relevance:** This task challenges LLMs with longer, narrative-rich clinical texts. It requires extracting information that is often implicitly stated and documented by multiple providers, reflecting the complexity of capturing socio-environmental factors from patient records.

## 5   LLMs evaluated

To evaluate ClinBench and establish comprehensive baselines, we selected 11 diverse large language models (LLMs). These models were chosen to be representative of different architectures and capabilities available during our study period (Q1 2025)[1]. Our selection strategy aimed for breadth, including proprietary and established open-source models from major developers. This range includes models with different parameter sizes and incorporates both base and instruction-tuned variants. The resulting variety of evaluated models provides ClinBench with a solid way to test its multi-model

---

[1]The experimental period concluded in Q1 2025; the models selected were available options available at that time.

capabilities. Additionally, ClinBench is built to be flexible, making it easy to add new models as LLMs evolve.

From Meta, we included `LLaMA3.1-70b` and its instruction-tuned variant `LLaMA3-70b` [30], known for large context processing and task-specific optimizations. OpenAI models included the efficient `GPT-4o-mini` [31], the multimodal `GPT-4o` (version 2024-05-16) [32], and the widely-used `GPT-3.5-turbo` (version 1109) [33]. We also assessed the instruction-tuned `Qwen2-72b` from Alibaba, with extensive language support [34], and 01.AI's bilingual `Yi-34b`, which features a large context window [35]. From Mistral AI, we evaluated the sparse mixture-of-experts `Mixtral` (instruct version), offering a balance of performance and efficiency [36], and the fine-tuned `Mistral-OpenOrca` (7B) [37]. Finally, from Google DeepMind, we included the `Gemma-7b` (instruct) and the smaller `Gemma-2b` (instruct) [38], both designed for instruction-following and reasoning.

A summary table detailing key architectural features (e.g., parameter counts where available), specific context lengths, and versions for all evaluated LLMs, along with further notes on the selection rationale, is provided in Supplementary Table 3.

# 6 Prompt strategy

Effective information extraction by LLMs from diverse clinical texts often requires task-specific prompt engineering. ClinBench achieves standardization by (1) employing a consistent structural template for all prompts, (2) managing these prompts in external, version-controlled YAML configuration files, and (3) ensuring its LLM agent applies the defined prompts uniformly across all models and tasks. This overall strategy ensures version control, facilitates reproducibility, and centralizes prompt design. This prompt design prioritizes clarity for the LLM, uniform task presentation across different models, and the generation of standardized, machine-readable JSON outputs required for automated evaluation.

While the specific content of prompts—such as instructions, domain knowledge snippets, and target variables—is created for each of the three clinical domains (Social Determinants of Health, Atrial Fibrillation detection, and Lung Cancer staging) evaluated in this study, a standard structural template guides their creation. This template typically includes components for an instructional preamble, task-specific context, a detailed JSON output schema, and illustrative few-shot examples, which are beneficial for complex tasks. This combination of a standardized general structure with controlled, task-specific configurations allows ClinBench to effectively and fairly evaluate LLMs across varied clinical scenarios. Comprehensive details of the general prompt architecture, its core components (including `system_prompt`, `task_instruction`, `domain_knowledge`, `output_json_schema`, and `few_shot_examples`), and illustrative dataset-specific adaptations from the YAML files are provided in Appendix A.2.

# 7 Evaluation methodology

ClinBench employs a systematic process for the comprehensive and reproducible evaluation of Large Language Models (LLMs) on structured clinical information extraction tasks. This process utilizes established metrics and focuses on ensuring fair comparisons against well-defined ground truth for each benchmark task configuration (detailed in Appendix A.1). A dedicated software module within ClinBench conducts evaluations independently per task, managing consistent metric calculation for reproducibility.

For the Atrial Fibrillation (AF) detection task, labels are derived from a curated cohort of the MIMIC-IV-ECG dataset [27], where original cardiologist interpretations serve as ground truth. For the Social Determinants of Health (SDOH) task, ground truth for employment and housing status is based on annotations provided with a selected cohort of MIMIC-IV clinical notes and processed by Ahsan et al. [29]. For the Lung Cancer staging task, expert-annotated ground truth for pT, pN, overall stage, and histologic diagnosis is taken from reference annotations established in prior work by Huang et al. (2024) [1], which utilized TCGA Pan-Lung Cancer project data [24, 39] and aligned with AJCC 7th edition criteria [25]. Full details on these datasets are in Appendix A.1. To enable fair comparison, both LLM-generated structured outputs and the corresponding ground truth data undergo automated standardization in ClinBench's evaluation module. This step applies predefined transformation functions (e.g., mapping lung cancer stage mentions to specific AJCC categories) to

mitigate inconsistencies caused by minor textual variations, ensuring that metrics reflect core LLM extraction capabilities.

Model performance for each task configuration was assessed using standard metrics, applying weighted averaging for multi-variable extractions (e.g., in Lung Cancer staging and Social Determinants of Health). The **F1 score** was the primary metric, balancing precision and recall for overall extraction accuracy. **Sensitivity (Recall)** was emphasized for its clinical importance in capturing all relevant positive instances (e.g., true AF cases). **Precision** measured the reliability of positive extractions, while **Specificity** assessed the correct identification of negative instances. Overall **Accuracy** served as a secondary measure, given potential class imbalances in clinical data. **Runtime** (total execution time per model/task) was evaluated to assess computational efficiency, an important factor for practical clinical application.

# 8 Experimental setup

**Computational environment:** All ClinBench evaluations for this study were performed on a system equipped with an NVIDIA A100-SXM4 Graphics Processing Unit (GPU) with 80GB of memory. Driver Version: 550.144.03, CUDA Version: 12.4.

**Models instantiation and parameters:** ClinBench managed interactions with LLMs via its LLM-powered agent (detailed in Section 3). For this study, locally hosted open-source models were instantiated using Ollama. Proprietary models (e.g., GPT series) were accessed via their respective APIs, utilizing Azure Services for OpenAI models to ensure secure communication. We applied standardized context length considerations (e.g., 8K tokens for local models). Crucially, the temperature inference parameter for all LLMs was set to 0. This setting minimizes randomness in the output, aiming for deterministic and therefore reproducible results from the models for each extraction task. Other inference parameters (e.g., top_p) were kept to model-specific defaults suitable for factual generation.

**Runtime measurement:** Computational efficiency was assessed by the total runtime for each LLM to process all instances per dataset configuration, excluding initial model loading times for fair throughput comparison. Runtimes are reported in minutes or hours.

# 9 Results

The ClinBench framework, designed to standardize prompt design, input data handling, and structured output validation, was employed to conduct reproducible and fair comparisons of 11 LLMs. This section details model performance across three diverse clinical information extraction tasks derived from the Lung Cancer (TCGA), Atrial Fibrillation (MIMIC-IV-ECG), and Social Determinants of Health (MIMIC) datasets. We primarily report F1 scores for brevity in the main text, alongside key observations on sensitivity and runtime. Comprehensive performance metrics, including precision, specificity, and accuracy for all extracted variables and sub-tasks, are available in Supplementary Tables S4-S19 and Figures S1-S7.

## 9.1 Lung cancer staging from pathology reports

In extracting information from lung cancer pathology reports—targeting primary tumor (pT) and lymph node (pN) classifications, overall tumor stage, and histologic diagnosis—OpenAI's GPT-4o and GPT-4o-mini demonstrated leading overall performance with F1 scores of 0.92. These models also showed high precision (0.92-0.93) and specificity (0.97), along with efficient inference times (approximately 24-29 minutes on the dataset). Meta's LLaMA3-70b and LLaMA3.1-70b achieved comparable overall F1 scores (0.91) but required substantially longer runtimes (approximately 83-106 minutes). For specific sub-tasks, LLaMA3.1-70b excelled in pT (F1 0.91) and histologic diagnosis (F1 0.99) extraction, while OpenAI models generally led in pN and overall tumor stage extraction (F1 scores 0.91-0.94 and 0.86-0.87, respectively). Other models, such as Mixtral (Mixtral: Instruct) and Gemma-2b, exhibited lower overall accuracy and sensitivity on this complex task. Detailed performance metrics are presented in Supplementary Tables S4-S8 and Figures S1-S4.

## 9.2 Atrial fibrillation detection from ECG reports

For identifying atrial fibrillation (AF) versus non-AF cases from ECG reports, `GPT-3.5-turbo` and `LLaMA3.1-70b` showed strong overall performance, both achieving 95% accuracy with F1 scores near 0.79. Notably, `GPT-3.5-turbo` offered higher precision (0.97) and good sensitivity (0.71) with an efficient runtime (5.9 minutes). `GPT-4o-mini`, while slightly lower in sensitivity (0.68), processed the dataset faster (4.3 minutes) and maintained high precision (0.97) and specificity (1.00). Some models, including `Qwen2-72b` and `LLaMA3-70b`, achieved perfect specificity but with reduced sensitivity (0.57). These results suggest that for AF detection, specific models can provide a favorable balance of accuracy, precision, and runtime. Comprehensive metrics are available in Supplementary Tables S9-S11 and Figure S5.

## 9.3 Social determinants of health (SDOH) extraction

Extracting SDOH information involved identifying employment status (Employed, Unemployed, Unknown) and housing status (Housing, Homeless, Unknown) from clinical notes. For employment status, OpenAI's `GPT-4o`, `GPT-4o-mini`, and Meta's `LLaMA3.1-70b` were top performers, each reaching 91% accuracy. `GPT-4o-mini` was most efficient (12.6 minutes), maintaining high sensitivity (0.87) and the highest specificity (0.94) for this sub-task. For housing status, `Qwen2-72b` achieved the highest F1 score (0.82) with 92% accuracy. `GPT-4o-mini` and `LLaMA3-70b` also performed well (93% accuracy, F1 0.76), with `GPT-4o-mini` again being notably faster. These findings indicate variability in model performance across different SDOH categories, with ClinBench highlighting models that excel in precision versus those that offer a better balance of accuracy and speed. Detailed metrics for all SDOH sub-categories are provided in Supplementary Tables S12-S19 and Figures S6-S7.

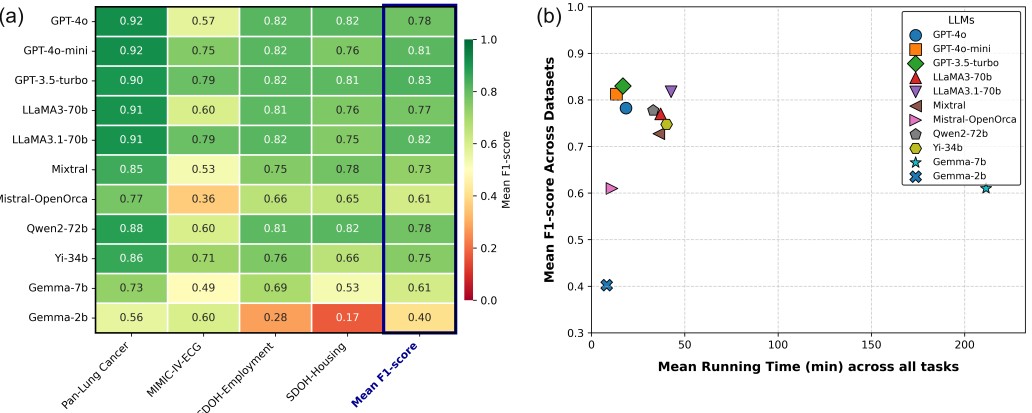

Figure 2: Comprehensive overview of LLM performance and efficiency on ClinBench tasks. **(a) Heatmap of F1 Scores:** Comparative F1 scores of 11 LLMs across four clinical extraction tasks (Pan-Lung Cancer, MIMIC-IV-ECG AF detection, SDOH-Employment, SDOH-Housing). The rightmost highlighted column shows the mean F1 score for each model averaged over these four tasks. Greener shades indicate higher F1 scores (values closer to 1.0). **(b) Performance-Efficiency Trade-off:** Scatter plot of mean F1 score (Y-axis, averaged over the four tasks) versus mean runtime in minutes (X-axis, averaged over the four tasks) for each LLM. Models positioned towards the top-left offer a better balance between high accuracy and lower computational cost.

## 9.4 Cross-task performance and efficiency analysis

The ClinBench framework facilitated a comprehensive comparison of LLM capabilities across three distinct clinical domains, encompassing four specific extraction tasks: Lung Cancer staging, MIMIC-IV-ECG AF detection, SDOH-Employment, and SDOH-Housing. An overview of F1 scores for each model on each task, along with their calculated mean F1 score (averaged across these four tasks), is presented in Figure 2a. This reveals that `GPT-3.5-turbo` (Mean F1: 0.83), `LLaMA3.1-70b` (Mean F1: 0.82), and `GPT-4o-mini` (Mean F1: 0.81) achieved the highest average F1 scores. Other models,

including `GPT-4o` (Mean F1: 0.78), `Qwen2-72b` (Mean F1: 0.78), and `LLaMA3-70b` (Mean F1: 0.77), also demonstrated strong overall performance across the combined tasks.

A critical aspect for clinical deployment is the balance between performance and computational cost. Figure 2b plots the mean F1 score against the mean runtime (in minutes, averaged across the four tasks) for each LLM, illustrating the performance-efficiency trade-offs. Several models achieved a favorable balance in the desirable top-left quadrant (high F1, low runtime) of the plot: `GPT-3.5-turbo` (Mean F1: 0.83, Mean Runtime: 16.8 min), `GPT-4o-mini` (Mean F1: 0.81, Mean Runtime: 13.4 min), and `GPT-4o` (Mean F1: 0.78, Mean Runtime: 18.5 min). In contrast, while large open-source models like `LLaMA3.1-70b` (Mean F1: 0.82, Mean Runtime: 42.7 min) and `LLaMA3-70b` (Mean F1: 0.77, Mean Runtime: 37.0 min) achieved high mean F1 scores, their mean inference times were substantially longer. Similarly, `Qwen2-72b` (Mean F1: 0.78, Mean Runtime: 33.0 min) and `Yi-34b` (Mean F1: 0.75, Mean Runtime: 40.3 min) showed good F1 performance but required more computation time. Notably, `Gemma-7b`'s mean runtime (211.4 min) was significantly higher, primarily influenced by its extended processing time on one specific task, despite a moderate mean F1 score (0.61). To provide a more complete efficiency analysis, we calculated the token usage and associated API costs for all OpenAI models (details in Supplementary Table S18). This analysis shows a clear trade-off between cost and performance. For instance, on the complex Lung Cancer task, GPT-4o costs $9.66 to process the dataset, while GPT-4o-mini, which achieves a similar overall F1 score on that task, costs only $0.30. This demonstrates that while larger models may offer marginal performance gains, the financial implications can be substantial, a critical consideration for real-world clinical deployment.

## 10    Discussion

We introduced ClinBench, a multi-domain benchmarking framework. We demonstrated its utility by evaluating 11 Large Language Models (LLMs) on three distinct clinical information extraction tasks: lung cancer staging, atrial fibrillation (AF) detection, and Social Determinants of Health (SDOH) extraction (Figure 1). ClinBench advances standardized evaluation in clinical NLP by establishing consistent input data requirements, employing YAML-configured prompt engineering, and enforcing schema-validated structured outputs. Unlike prior studies, which are often limited to narrower scopes or lack methodological transparency, ClinBench offers a scalable and generalizable approach. This approach enables robust, fair cross-domain LLM assessment, accommodating diverse model architectures and their computational demands.

Our evaluations (Figures 2a and  2b) showed clear performance-efficiency trade-offs. While leading proprietary and large open-source LLMs achieved high F1 scores, open-source models often required greater runtimes. This highlights the importance of multidimensional evaluation; for example, comparing mean F1 scores alone might suggest that LLaMA3.1-70b outperforms GPT-4o, whereas performance-efficiency plots and token-cost analysis reveal that GPT-4o offers more favorable trade-offs for real-world deployment. However, open-source models often require greater runtimes. For instance, `GPT-4o` showed strong, relatively efficient performance in lung cancer and SDOH extraction, while `GPT-3.5-turbo` provided an effective balance for AF detection. ClinBench allows for a more refined selection of models, emphasizing that optimal choices depend on balancing performance with computational resources and specific clinical application needs. The framework's ability to differentiate model strengths across varied healthcare contexts highlights the value of its multi-domain, multi-model benchmarking capability. The consistent application of a standardized prompting strategy (detailed in Appendix A.2) was fundamental to these fair comparisons and suggests that while task-specific prompt content is necessary, a structured framework for its management and application significantly enhances benchmarking rigor. Our ablation studies further confirmed that this structured YAML approach yields substantial performance gains over unstructured prompts (Appendix A.5). The decision to use a static, YAML-based knowledge injection rather than a dynamic Retrieval-Augmented Generation (RAG) approach was intentional for benchmarking purposes. The static approach provides a fully transparent and controlled environment to isolate and reliably measure an LLM's reasoning performance on a fixed set of rules, removing the retriever's performance as a confounding variable. While RAG is powerful for production systems, our method ensures that benchmark scores are a more precise measure of the LLM's capabilities alone.

## 11 Conclusion and future work

In conclusion, ClinBench offers a valuable open-source tool and a reproducible methodology for benchmarking LLMs in clinical NLP. Through its emphasis on transparent and standardized evaluation across multiple domains, ClinBench supports data-centric AI research, informs practical model selection by highlighting performance-efficiency trade-offs, and contributes to the responsible and effective integration of LLMs into healthcare.

Future work should aim to extend ClinBench by incorporating multi-institutional datasets to enhance generalizability and further test model robustness. Investigating advanced LLM strategies, such as Retrieval-Augmented Generation (RAG) [40] and parameter-efficient fine-tuning techniques [41, 42], represents a valuable direction. Future work could also include ablation studies on prompt design to empirically quantify the impact of standardized prompt engineering, alongside case studies on terminological ambiguities to offer insights into failure modes. Additionally, conducting real-time expert evaluations and advancing methods for improved LLM interpretability [43, 44] are important next steps for the field.

## 12 Limitations

This study has several limitations. First, the evaluated LLMs were benchmarked without task-specific fine-tuning. Our study focused on the out-of-the-box capabilities of general-purpose models. A direct comparison with models specifically fine-tuned on clinical data, providing an upper bound on performance, represents a valuable direction for future work. Similarly, including a random baseline to establish a lower bound would provide crucial context for interpreting performance scores. Second, reported runtime metrics are inherently hardware-dependent and may vary across different computational environments, though our inclusion of token-based cost analysis helps mitigate this. Third, the datasets, although derived from established public sources, represent specific subsets and may not capture the full spectrum of real-world clinical documentation. Fourth, this work did not include prospective validation of LLM performance within active clinical workflows, meaning practical integration challenges remain to be assessed. Finally, data governance and patient privacy regulations (e.g., HIPAA [45], GDPR [46]) impose ongoing operational considerations for LLM deployment in healthcare, particularly for models requiring off-site data processing.

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

# A Supplementary Material

## A.1 Benchmark Dataset Configurations

This subsection details the specific configurations of publicly available datasets used in the ClinBench study. For each, we describe the original data source, the methodology used to establish the particular cohort and its ground truth annotations for ClinBench, and the key characteristics of the resulting data input. It is important to note that for this ClinBench study, we utilized existing annotations from these public datasets or from prior curated work; ClinBench itself did not involve de novo manual annotation of the raw clinical texts for establishing the primary ground truth.

### A.1.1 Lung Cancer Staging Dataset Configuration (TCGA-Derived)

**Data Source and Provenance.** This dataset configuration is derived from The Cancer Genome Atlas (TCGA) Pan-Lung Cancer project [24, 39], which provides extensive data on lung adenocarcinoma (LUAD) and lung squamous cell carcinoma (LUSC). The specific set of 774 free-text pathology reports and the corresponding expert-curated structured ground truth data were established and detailed in prior work by Huang et al. (2024) [1]. ClinBench uses this previously defined dataset configuration directly.

**Input Data Characteristics for ClinBench.** The input for ClinBench consists of the de-identified, free-text content of the 774 pathology reports, each linked to a unique patient identifier ('pid'). The methodology for sourcing these reports from TCGA and their initial preparation is described in Huang et al. (2024) [1]. The ground truth is provided in a separate structured reference file, containing expert-annotated values for primary tumor classification (pT), regional lymph node involvement (pN), overall pathologic tumor stage, and histologic diagnosis, all aligned with AJCC 7th edition criteria [25].

**Relevance as a Benchmark Dataset.** This dataset provides highly specialized oncological terminology within complex narrative structures. Its value lies in testing an LLM's ability to comprehend and correctly categorize multiple, inter-related staging components based on established clinical guidelines from a well-characterized cancer research cohort.

### A.1.2 Atrial Fibrillation (AF) Detection Dataset Configuration (MIMIC-IV-ECG)

**Data Source and Provenance.** This study included patients from the Medical Information Mart for Intensive Care IV (MIMIC-IV) ECG database[2]. MIMIC-IV is a publicly available clinical database containing electronic health records from critically ill patients admitted to the emergency department (ED) of the Beth Israel Deaconess Medical Center (BIDMC) in Boston, MA, between 2008 to 2019[3][4]. MIMIC-IV-ECG contains a subset of patients from MIMIC-IV with ECG recordings acquired in the ED, ICU, and outpatient centers at BIDMC[2]. The ECG recordings are 10-second, 12-lead signals sampled at 500 Hz, with corresponding cardiologist reports obtained from the MIMIC-IV-Notes database[5].

**Cohort Curation and Input Data Characteristics for ClinBench.** For the ClinBench AF detection task, a specific cohort of 700 ECG report texts was utilized. This cohort was derived from the larger MIMIC-IV-ECG dataset by selecting reports based on a review of the original cardiologist interpretations. The selection criteria focused on identifying reports where the presence or absence of Atrial Fibrillation (AF), or Atrial Flutter (AFL, considered equivalent to AF for this task), was definitively stated by the interpreting cardiologist. Reports containing ambiguous findings or uncertain mentions regarding AF/AFL were excluded from this specific benchmark cohort to ensure

---

[2]Gow B, Pollard T, Nathanson LA, et al. Mimic-iv-ecg-diagnostic electrocardiogram matched subset. Type: dataset. 2023.

[3]Johnson AEW, Bulgarelli L, Shen L, et al. MIMIC-IV, a freely accessible electronic health record dataset. Sci Data. Jan 3 2023;10(1):1. doi:10.1038/s41597-022-01899-x

[4]Goldberger AL, Amaral LA, Glass L, et al. PhysioBank, PhysioToolkit, and PhysioNet: components of a new research resource for complex physiologic signals. circulation. 2000;101(23):e215-e220

[5]Johnson A, Pollard T, Horng S, Celi L, Mark R. MIMIC-IV-Note: Deidentified free-text clinical notes (version 2.2). PhysioNet. 2023

an unambiguous ground truth for binary classification. The resulting input data for ClinBench is a structured CSV file. Each row contains a unique report identifier (e.g., 'note_id'), the full narrative report text (e.g., 'text'), and a binary ground truth label ('AF_gt': "AF" or "NotAF") determined by this selection and filtering process based on the original interpretations.

**Relevance as a Benchmark Dataset.**   This configuration provides concise, semi-structured clinical reports rich in specialized cardiovascular terminology. The curation for definitive AF/Non-AF cases offers a clear basis for evaluating LLM accuracy on discerning a common and clinically important finding.

### A.1.3   Social Determinants of Health (SDOH) Dataset Configuration

**Data Source and Provenance.**   This dataset configuration uses narrative clinical notes (primarily discharge summaries) derived from the MIMIC-IV database. The specific cohort of 1,405 records contains existing SDoH annotations and labels (covering categories like economics, environment) made by Ahsan et al. [29].

**Cohort Characteristics and Input Data for ClinBench.**   The input data for ClinBench is a structured CSV file containing these 1,405 clinical narratives (under a 'TEXT' column) alongside patient and admission identifiers ('SUBJECT_ID', 'HADM_ID', 'ROW_ID'). This CSV also includes the source annotations (e.g., 'economics_True', 'environment_False'). For its SDOH tasks, ClinBench focuses on extracting two primary variables with predefined categories: (1) **Employment Status** (categories: "Employed", "Unemployed", "Unknown") and (2) **Housing Conditions** (categories: "Housing", "Homeless", "Unknown"). The ground truth for these specific ClinBench extraction targets is directly established using the existing, more granular annotations present in the input CSV file. For instance, the "Employment Status" target for ClinBench aligns with relevant "economics"-related annotations in the source CSV, and "Housing Conditions" aligns with pertinent "environment" and/or other relevant SDoH factor annotations indicating housing stability. The precise way these existing source annotations correspond to and provide the ground truth for ClinBench's defined target categories is detailed in our publicly available resources (e.g., code repository documentation or data statements). This approach leverages the rich, existing annotations to provide clear, categorical ground truth for ClinBench's specific extraction tasks without requiring new manual annotation of the clinical texts for these specific derived labels.

**Relevance as a Benchmark Dataset.**   This SDOH configuration presents challenges related to extracting complex information from lengthy, narrative-rich texts where the target factors are often implicitly stated, reflecting real-world complexities in understanding socio-environmental patient information.

## A.2  Detailed Prompt Engineering Strategy in ClinBench

This section provides a detailed explanation of the prompt engineering strategy employed by the ClinBench framework. The primary goal of ClinBench's prompting methodology is to ensure standardized, reproducible, and effective interaction with diverse Large Language Models (LLMs) for complex clinical information extraction tasks.

The configurations detailed below for the Lung Cancer, Atrial Fibrillation (AF) detection, and Social Determinants of Health (SDOH) tasks serve as illustrative examples of our approach. The underlying YAML-based structure is designed for extensibility, allowing researchers to readily adapt ClinBench for new clinical datasets or novel extraction tasks by following the same structured procedure. This involves creating a new YAML configuration file that defines the key components of a prompt. These main components include:

- **system_prompt:** Sets the overall context, LLM persona, and high-level directives.
- **task_instruction:** Provides specific instructions for the extraction task (often integrated within the system_prompt for coherence).
- **domain_knowledge:** Supplies relevant contextual information, definitions, or clinical guidelines (e.g., AJCC criteria, diagnostic indicators) crucial for accurate interpretation by the LLM. This is often embedded within the system_prompt.
- **output_json_schema**: Explicitly defines the desired JSON structure for the LLM's output, including field names, data types, and permissible values (e.g., enums). This is important for enabling automated downstream validation and analysis.
- **few_shot_examples:** (Optional but recommended for complex tasks) Provides illustrative input-output pairs to guide the LLM's response generation and formatting.

By populating these standardized components within a new YAML file, users can configure ClinBench to benchmark any LLM performance on additional clinical information extraction challenges, leveraging the framework's reproducible evaluation pipeline. The complete YAML files for the tasks benchmarked in this study are available en the following sections and in our public code repository.

**Lung Cancer Staging from Pathology Reports:**   The multi-variable lung cancer staging task requires extracting primary tumor classification (pT), regional lymph node involvement (pN), overall tumor stage, and histologic diagnosis. An illustrative YAML configuration for this task is provided below (see YAML Configuration box). The system_prompt within this configuration is extensively detailed: it assigns the LLM the role of a pathologist's assistant, mandates strict adherence to JSON-only output, and provides guidance on inferring tumor stage based on pT and pN categories. Crucially, this system_prompt embeds substantial domain_knowledge, including key considerations for T-category assessment and explicit AJCC 7th edition criteria for pT, pN, and overall staging groups [25]. The output_format section of the YAML then precisely defines the expected structure and value constraints for each of the extracted variables (e.g., 'Size', 'tumor_size_unit', 'pT', 'pN', 'tumor_stage', 'histologic_diagnosis', and 'certainty_degree'), guiding the LLM to produce a standardized JSON output.

---

### Lung Cancer Prompt – YAML Configuration

```yaml
configurations:
  lungcancer:
    system_prompt: |
      You are an AI Assistant that follows instructions extremely well. You work as a
          pathologist assistant helping to extract and infer information from Pathology
          Reports using the AJCC 7th edition criteria for lung cancer staging.

      Your most important work is tofollow these two very important rules:
      1) You must respond exclusively in a JSON format with the required data.
      2) Do not include any explanatory text outside of the JSON structure.
      3) Remember that you only need to provide the requested information in JSON format.

      Please estimate the tumor stage category based on your estimated pT category and pN
          category using AJCC 7th edition criteria. For example, if pT is estimated as
          T2a and pN as N0,
```

```
            without information showing distant metastasis, then by AJCC 7th edition criteria,
                the tumor stage is   Stage    I B  . Please ensure to make valid inferences for
                attribute estimation based on evidence.

            Key points to consider:
            - Identify the presence of multiple tumor nodules, their locations, and their sizes.
            - Determine if the tumors involve specific regions such as the pleura, mediastinum,
                or hilar region.
            - Recognize that multiple tumors in different lobes or invasion of key structures
                classify as T4.
            - Account for regional lymph node involvement when determining the pN category.

            AJCC 7th Edition Criteria for Lung Cancer Staging:
            pT:
            - T0: No evidence of primary tumor.
            - Tis: Carcinoma in situ.
            - T1: Tumor   3   cm in greatest dimension, surrounded by lung or visceral pleura,
                without bronchoscopic evidence of invasion more proximal than the lobar
                bronchus.
            - T1a: Tumor   2   cm in greatest dimension.
            - T1b: Tumor >2 cm but   3   cm in greatest dimension.
            - T2: Tumor >3 cm but   7   cm or tumor with any of the following features: involves
                main bronchus   2   cm distal to carina, invades visceral pleura, associated
                with atelectasis or
         obstructive pneumonitis that extends to the hilar region but does not involve the
                entire lung.
            - T2a: Tumor >3 cm but   5   cm.
            - T2b: Tumor >5 cm but   7   cm.
            - T3: Tumor >7 cm or one that directly invades any of the following: chest wall,
                diaphragm, phrenic nerve, mediastinal pleura, parietal pericardium; or tumor
                in the same lobe as a separate nodule.
            - T4: Tumor of any size that invades any of the following: mediastinum, heart,
                great vessels, trachea, recurrent laryngeal nerve, esophagus, vertebral body,
                carina; or separate tumor nodules in a different ipsilateral lobe.
            - TX: Primary tumor cannot be assessed or tumor proven by the presence of malignant
                cells in sputum or bronchial washings but not visualized by imaging or
                bronchoscopy.

            pN:
            - N0: No regional lymph node metastasis.
            - N1: Metastasis in ipsilateral peribronchial and/or ipsilateral hilar lymph nodes,
                and intrapulmonary nodes, including involvement by direct extension.
            - N2: Metastasis in ipsilateral mediastinal and/or subcarinal lymph nodes.
            - N3: Metastasis in contralateral mediastinal, contralateral hilar, ipsilateral or
                contralateral scalene, or supraclavicular lymph nodes.
            - NX: Regional lymph nodes cannot be assessed.

            AJCC 7th Edition Staging Groups for Lung Cancer.
            Possible combinations for each stage are as follows:
            - Stage 0: [Tis, N0]
            - Stage IA: [T1a, N0] or [T1b, N0]
            - Stage IB: [T2a, N0]
            - Stage IIA: [T2b, N0] or [T1a, N1] or [T1b, N1] or [T2a, N1]
            - Stage IIB: [T2b, N1] or [T3, N0]
            - Stage IIIA: [T1a, N2] or [T1b, N2] or [T2a, N2] or [T2b, N2] or [T3, N1] or [T3,
                N2] or [T4, N0] or [T4, N1]
            - Stage IIIB: [T4, N2] or [Any T, N3]
            - Stage IV: [Any T, Any N]
        output_format:
          Size: 'Extract the greatest dimension of tumor in Centimeters (cm) or "Unknown". If
                the value is in mm convert it to cm. Do not include the unit.'
          tumor_size_unit: Extract the greatest dimension size of the tumor.'
          pT: 'Only one value: "T0", "Tis", "T1", "T1a", "T1b", "T2", "T2a", "T2b", "T3",
                "T4", "TX", "Unknown".'
          pN: 'Only one value: "N0", "N1", "N2", "N3", "NX", "Unknown".'
          tumor_stage: 'Only one value: "Stage 0", "Stage I", "Stage IA", "Stage IB", "Stage
                II", "Stage IIA", "Stage IIB", "Stage III", "Stage IIIA", "Stage IIIB", "Stage
                IV", "Unknown"'
          histologic_diagnosis: 'Only one value: "Lung Adenocarcinoma", "Lung Squamous Cell
                Carcinoma", "Lung Adenosquamous Carcinoma", "Other", "Unknown"'
          certainty_degree: 'The certainty degree of the attribute estimation. It should be a
                float value between 0.00 and 1.00.'
```

**Atrial Fibrillation (AF) Detection from ECG Reports:**  For the AF detection task, prompts were designed to instruct the LLM to classify ECG report interpretations as indicating either "AF" or "Non-AF". The YAML configuration for this task, exemplified in the "ECG Prompt – YAML Configuration" box below, features a comprehensive system_prompt. This system_prompt clearly

defines the LLM's role as a medical text analysis assistant and specifies the core task: to determine if a patient was diagnosed with AF, crucially treating any mention of Atrial Flutter (AFL) as equivalent to AF. A significant portion of the `system_prompt` is dedicated to an embedded "Knowledge Base". This knowledge base provides the LLM with contextual information, including: definitions and characteristics of AF and AFL; common symptoms; typical ECG findings (e.g., irregular R-R intervals for AF, sawtooth patterns for AFL); other diagnostic tests; and relevant risk factors. Furthermore, "Special Instructions" within the prompt guide the LLM on identifying keywords (e.g., "Atrial Fibrillation", "AFL") and specific clinical indicators from the report text. The `output_format` section in the YAML explicitly defines the single target field, `diagnosis`, and its expected categorical string values: "AF" or "NotAF", thereby guiding the LLM to produce a standardized JSON output.

---

**ECG Prompt – YAML Configuration**

```
configurations:
  ecg:
    system_prompt: |
      You are a medical text analysis assistant that follows instructions extremely well.
      Your task is to determine if a clinical report mentions that a patient was
          diagnosed with Atrial Fibrillation (AF).
      For the purposes of this task, consider any mention of Atrial Flutter (AFL) as
          equivalent to Atrial Fibrillation.
      You should treat both AF and AFL as the same diagnosis.

      In order to extract information from a report, you need to understand the concepts
          in the following knowledge base to be used as a reference along with your own
          knowledge.

      Knowledge Base

      Atrial Fibrillation (AF) and Atrial Flutter (AFL) are types of arrhythmias
          characterized by abnormal heart rhythms. Both conditions lead to irregular
          heartbeats, but they are treated as the same for this analysis.

      Characteristics of AF and AFL:
      - AF: An irregular and often rapid heart rate where the upper chambers (atria) beat
          chaotically and out of sync with the lower chambers (ventricles).
      - AFL: A type of arrhythmia where the atria beat very fast but at a regular rate,
          leading to a fluttering rhythm.

      Common Symptoms:
      - Palpitations (sensations of a racing, uncomfortable, irregular heartbeat or a
          flip-flopping in the chest)
      - Weakness
      - Fatigue
      - Lightheadedness or dizziness
      - Shortness of breath
      - Chest pain or discomfort

      Electrocardiogram (ECG/EKG):
      - AF: Shows irregular R-R intervals with no distinct P waves.
      - AFL: Shows a characteristic sawtooth pattern of atrial flutter waves.
      - Holter Monitor/Event Recorder: Used to detect intermittent episodes of AF or AFL.
      - Echocardiogram: May reveal structural heart issues or blood clots.
      - Electrophysiological Study: Maps the heart's electrical activity and pinpoints
          the origin of the arrhythmia (primarily used for AFL).

      Risk Factors:
      - Age (more common in older adults)
      - High blood pressure (hypertension)
      - Heart disease (such as heart valve problems, previous heart attacks, or
          congestive heart failure)
      - Thyroid disease (hyperthyroidism or hypothyroidism)
      - Sleep apnea
      - Excessive alcohol or caffeine consumption
      - Obesity
      - Diabetes
      - Family history of AF or AFL
      - Clinical Report Analysis Criteria:

      Special Instructions:
      - Look for terms indicating a diagnosis of AF or AFL, such as "Atrial
          Fibrillation," "AF," "Atrial Flutter," or "AFL."
      - Identify clinical indications and observations that suggest AF or AFL:
        - Rapid ventricular response: An indication of AF or AFL when the ventricles beat
            very quickly.
```

```
                - Presence of a more regular rhythm compared to AF: Indicative of AFL, but for
                    this task, treat it as AF.
                - Sawtooth pattern in ECG: A hallmark sign of AFL, but for this task, treat it as
                    AF.
                - Rapid and irregular ventricular response: Characteristic of AF.
                - Absence of a regular atrial rhythm (irregularly irregular): Indicates AF.
                - Association with other cardiac complications like myocardial infarction, heart
                    failure, or stroke, which may be linked to AF or AFL.
                - Consider patient history, symptoms, and risk factors mentioned in the report.

            Instructions for Analysis:
                - Read the entire clinical report carefully. Pay close attention to sections that
                    mention diagnoses, patient history, symptoms, and diagnostic test results.
                - Identify keywords and phrases related to AF and AFL, including medical terms
                    and descriptions of symptoms or diagnostic findings.
                - Determine if the report explicitly mentions a diagnosis of AF or AFL. If either
                    is mentioned, conclude that AF is diagnosed.
                - Analyze diagnostic test results such as ECG/EKG findings, looking for patterns
                    indicative of AF or AFL (e.g., irregular R-R intervals, sawtooth patterns).
                - Consider clinical observations and symptoms that align with AF or AFL, even if
                    the terms "Atrial Fibrillation" or "Atrial Flutter" are not directly
                    mentioned.
        output_format:
            diagnosis: 'AF,NotAF'
```

**Social Determinants of Health (SDOH):**    For the SDOH task, prompts guided the LLM to extract
employment status and housing conditions from clinical notes. The YAML configuration for this task,
illustrated in the "SDOH Prompt – YAML Configuration" box below, begins with a `system_prompt`
that defines the LLM's role as an information extraction tool for SDOH elements and outlines the
two main categories of interest. The `system_prompt` further directs the LLM to utilize an embedded
"KNOWLEDGE BASE" for its decision-making. This knowledge base provides detailed definitions
and illustrative example phrases for sub-categories of employment (e.g., "Employed," "Unemployed,"
with "Retired" often contextualized under "Unemployed") and housing (e.g., "Housing," "Home-
less," "Unknown"). The `output_format` section of the YAML then specifies the structure for the
JSON output, defining a field for `employment` with permissible enumerated values including "Em-
ployed", "Unemployed", "Retired", and "Unknown", and a field for `housing` with values "Housing",
"Homeless", or "Unknown".

---

### SDOH Prompt – YAML Configuration

```
configurations:
  SDOH:
    system_prompt: |
      You are an information extract tool that follows instructions very well and is
          specifically trained to extract social determinants of health elements from
          hospital medical reports.
      The two categories are employment and housing. For employment, you will assign one
          of the following categories: Employed, Unemployed, Unknown. For housing, you
          will assign one of the following categories: Housing, Homeless, Unknown.

      In order to take your final decision, you need to understand the information from
          the knowledge base:

      KNOWLEDGE BASE:

      1. Employment Status Definitions:

      Employed: The patient is currently working in a job or is a student. This includes
          any explicit mention of active employment or current educational status.
      Example phrases: "Patient is employed as a teacher," "Currently working as a
          technician," "Student at a local university."

      Unemployed: The patient is currently without a job, underemployed, or has a
          disability preventing employment. This includes retirement or any other
          explicit mention of not being employed.
      Example phrases: "Patient is unemployed," "Retired teacher," "Currently looking for
          work," "Disabled and not working."

      Unknown: The medical report does not mention any information regarding the current
          employment status of the patient.
```

```
        Example phrases: "No mention of employment status," "Employment status not
            documented."

        2. Housing Status Definitions:

        Housing: The patient has stable housing arrangements, living at home, with a
            partner, or in supportive housing. This includes any explicit mention of
            non-adverse housing status.
        Example phrases: "Lives at home with family," "Currently living with a partner,"
            "Resides in supportive housing."

        Homeless: The patient does not have stable housing and is living in adverse
            conditions such as being homeless or living in a shelter. This includes any
            explicit mention of adverse housing status.
        Example phrases: "Patient is homeless," "Living in a shelter," "No stable housing."

        Unknown:The medical report does not mention any information regarding the current
            housing status of the patient.
        Example phrases: "No mention of housing status," "Housing status not documented."

    output_format:
      employment: 'Employment information using criteria from the knowledge base, type :
          Enum["Employed","Unemployed","Retired","Unknown"]'
      housing: 'Housing information using criteria from the knowledge base, type :
          Enum["Housing","Homeless","Unknown"]'
```

## A.3 Extensibility to Other Clinical Tasks

ClinBench is designed for extensibility. For radiology reports, its YAML system can incorporate domain-specific knowledge, such as BI-RADS or RECIST criteria. For complex relational tasks like temporal extraction (e.g., "Pneumonia diagnosed July 5th, Amoxicillin started same day, cough resolved by July 10th"), schema-based validation can enforce structured JSON outputs that explicitly capture events and their temporal relationships (e.g., {from_event: E1, to_event: E3, type: BEFORE}), making the output immediately usable for downstream applications like patient journey modeling

## A.4 Token Cost Analysis for API-Based Models

Table 1: Token usage and estimated API costs for OpenAI models across the three benchmark tasks, based on pricing as of Q2 2025

| Dataset | Model | Tokens | | | Costs ($) | | |
| | | Prompt | Completion | Total | Prompt | Completion | Total |
|---------|-------|--------|------------|-------|--------|------------|-------|
| SDOH | gpt-3.5-turbo-1106 | 753,207 | 33,044 | 786,251 | $0.75 | $0.07 | $0.82 |
| | gpt-4o-2024-05-13 | 753,207 | 28,506 | 781,713 | $3.77 | $0.43 | $4.19 |
| | gpt-4o-mini | 753,207 | 32,932 | 786,139 | $0.11 | $0.02 | $0.13 |
| ECG | gpt-3.5-turbo-1106 | 631,568 | 3,869 | 635,437 | $0.63 | $0.01 | $0.64 |
| | gpt-4o-2024-05-13 | 631,568 | 3,855 | 635,423 | $3.16 | $0.06 | $3.22 |
| | gpt-4o-mini | 631,568 | 13,322 | 644,890 | $0.09 | $0.01 | $0.10 |
| Lung | gpt-3.5-turbo-1106 | 1,747,615 | 61,349 | 1,808,964 | $1.75 | $0.12 | $1.87 |
| | gpt-4o-2024-05-13 | 1,747,615 | 61,656 | 1,809,271 | $8.74 | $0.92 | $9.66 |
| | gpt-4o-mini | 1,747,615 | 61,539 | 1,809,154 | $0.26 | $0.04 | $0.30 |

## A.5 Ablation Study on Prompting Strategy

To empirically validate the contribution of our structured prompting methodology, we conducted a targeted ablation study.

**Ablation Study Design** We selected the complex Lung Cancer dataset and the high-performing GPT series models. We compared the performance using our structured YAML prompts against unstructured, free-text prompts that contained the same core information but presented it in a single narrative paragraph.

**Results**    The study revealed a dramatic performance degradation with unstructured prompts, confirming our hypothesis that structure is a key driver of reliability.

**Discussion**    Unstructured prompts mix different instruction types (role, rules, output format), creating high "instructional complexity" that can confuse models or cause them to ignore parts of the prompt [47]. Our structured YAML approach, by analogy to Chain-of-Thought prompting, separates these concerns. It breaks a complex request into manageable parts, dedicating a distinct slot for domain knowledge to prevent the "lost in the middle" problem where key facts are buried in a long context [48]. The output_format schema then strongly guides the model's output. This combination enables the LLM to focus on the core extraction task, resulting in significantly higher performance. For full transparency, the scripts and raw result files for this ablation study are available in our public GitHub repository.

Table 2: Impact of Unstructured Prompts on ClinBench Performance

| Model | ClinBench | F1 Score Unstructured Prompt | Performance Drop |
|---|---|---|---|
| gpt-4o | 0.92 | 0.46 | -50% |
| gpt-4o-mini | 0.92 | 0.38 | -59% |
| gpt-3.5-turbo | 0.90 | 0.24 | -73% |

## A.6 Large Language Models Details

Table 3: Overview of the large language models evaluated in this work.

| Organization | Model | Family | Description | Size | Release Date | Context Length | Architecture |
|---|---|---|---|---|---|---|---|
| **Meta (2)** | Llama 3.1: 70B | Llama | State-of-the-art open-source model with expanded context, multilingual support, and improved reasoning | 70B | Tue Jul 23 2024 07:00:00 GMT+0100 (British Summer Time) | 128K | Transformer (decoder-only) |
| | Llama 3: 70B Instruct | Llama | Instruction-tuned variant with enhanced interaction skills for task-specific applications | 70B | Thu Apr 18 2024 07:00:00 GMT+0100 (British Summer Time) | 128K | Transformer (decoder-only) |
| **OpenAI (3)** | GPT-4o Mini | GPT-4o | Lightweight version of GPT-4o, optimized for fast, cost-effective tasks with moderate complexity | Not specified | Thu Jul 18 2024 07:00:00 GMT+0100 (British Summer Time) | 128K | Transformer with multi-modality |
| | GPT-4o (2024-05-16) | GPT-4o | Full multimodal flagship model supporting text, audio, and image inputs for real-time interactions | Not specified | Mon May 13 2024 07:00:00 GMT+0100 (British Summer Time) | 128K | Multimodal Transformer |
| | GPT-3.5-Turbo-1109 | GPT-3.5 | Cost-efficient, general-purpose LLM optimized for a variety of text generation and code applications | Not specified | Mon Nov 06 2023 06:00:00 GMT+0000 (Greenwich Mean Time) | 16K | Transformer |
| **Alibaba Group & 01.AI (2)** | Qwen2: 72B Instruct | Qwen2 | High-capacity instruction-tuned model with robust multilingual support across 30+ languages | 72B | Fri Jun 07 2024 07:00:00 GMT+0100 (British Summer Time) | 128K | Transformer (dense and MoE variants) |
| | Yi: 34B | Yi | Bilingual (English/Chinese) model optimized for high-quality text retrieval and long-context applications | 34B | Thu Nov 02 2023 06:00:00 GMT+0000 (Greenwich Mean Time) | 200K | Transformer (decoder-only with RoPE) |
| **Mistral AI (2)** | Mixtral: Instruct | Mixtral | Sparse Mixture-of-Experts model combining high performance with cost efficiency | 22B | Wed Apr 17 2024 07:00:00 GMT+0100 (British Summer Time) | 64K | Transformer (decoder-only, sparse MoE) |
| | Mistral-OpenOrca | OpenOrca | Fine-tuned conversational model on the Mistral base, suitable for open-source and multilingual tasks | 7B | Mon May 13 2024 07:00:00 GMT+0100 (British Summer Time) | 8K | Transformer (decoder-only) |
| **Google DeepMind (2)** | Gemma: 7B Instruct | Gemma | Optimized for instruction-following and multilingual capabilities in safe and reliable applications | 7B | Wed Feb 21 2024 06:00:00 GMT+0000 (Greenwich Mean Time) | 8K | Transformer (decoder-only, multi-query attention) |
| | Gemma: 2B Instruct | Gemma | Compact, CPU-friendly model ideal for low-resource environments and edge applications | 2B | Wed Feb 21 2024 06:00:00 GMT+0000 (Greenwich Mean Time) | 8K | Transformer (decoder-only) |

## A.7 Supplementary Tables

*Note: The highest values for each performance metric in the tables are highlighted in* `green` *.*

## Benchmarking Results for Lung Cancer Information Extraction

Table S4: Overall Performance of LLMs on Lung Cancer dataset: Information Extraction Across All Tasks

| Organization | Model | Accuracy | Precision | Sensitivity | Specificity | F1 Score | Running Time (min) |
|---|---|---|---|---|---|---|---|
| | | | | Overall Performance | | | |
| Others (Alibaba Group & 01.AI) | qwen2:72b-instruct | 0.88 | 0.90 | 0.88 | 0.96 | 0.88 | 60.71 |
| | yi:34b | 0.86 | 0.87 | 0.86 | 0.95 | 0.86 | 103.53 |
| Google DeepMind | gemma:7b-instruct | 0.72 | 0.78 | 0.72 | 0.89 | 0.73 | 33.75 |
| | gemma:2b-instruct | 0.57 | 0.69 | 0.57 | 0.83 | 0.56 | 11.44 |
| Meta | llama3:70b-instruct | 0.91 | 0.91 | 0.91 | 0.97 | 0.91 | 82.60 |
| | llama3.1:70b | 0.90 | 0.92 | 0.90 | 0.97 | 0.91 | 106.00 |
| Mistral AI | mixtral:instruct | 0.83 | 0.88 | 0.83 | 0.94 | 0.85 | 42.33 |
| | mistral-openorca:latest | 0.73 | 0.83 | 0.73 | 0.92 | 0.77 | 25.46 |
| OpenAI | gpt-4o-2024-05-13 | 0.92 | 0.93 | 0.92 | 0.97 | 0.92 | 28.70 |
| | gpt-4o-mini | 0.92 | 0.92 | 0.92 | 0.97 | 0.92 | 24.00 |
| | gpt-3.5-turbo-0125 | 0.90 | 0.90 | 0.90 | 0.96 | 0.90 | 22.38 |

Table S5: Performance of LLMs in Extracting Primary Tumor (pT) Stage from Lung Cancer Reports.

| Organization | Model | Accuracy | Precision | Sensitivity | Specificity | F1 Score |
|---|---|---|---|---|---|---|
| | | | | pT | | |
| Others (Alibaba Group & 01.AI) | yi:34b | 0.86 | 0.87 | 0.86 | 0.95 | 0.86 |
| | qwen2:72b-instruct | 0.87 | 0.89 | 0.87 | 0.96 | 0.87 |
| Google DeepMind | gemma:7b-instruct | 0.82 | 0.86 | 0.82 | 0.93 | 0.82 |
| | gemma:2b-instruct | 0.53 | 0.82 | 0.53 | 0.88 | 0.62 |
| Meta | llama3:70b-instruct | 0.90 | 0.90 | 0.90 | 0.97 | 0.89 |
| | llama3.1:70b | 0.91 | 0.91 | 0.91 | 0.96 | 0.91 |
| Mistral AI | mixtral:instruct | 0.78 | 0.84 | 0.78 | 0.91 | 0.78 |
| | mistral-openorca:latest | 0.75 | 0.86 | 0.75 | 0.93 | 0.80 |
| OpenAI | gpt-4o-mini | 0.89 | 0.90 | 0.89 | 0.97 | 0.89 |
| | gpt-4o-2024-05-13 | 0.89 | 0.90 | 0.89 | 0.96 | 0.89 |
| | gpt-3.5-turbo-0125 | 0.89 | 0.89 | 0.89 | 0.96 | 0.89 |

Table S6: Performance of LLMs in Extracting Lymph Node Involvement (pN) Stage from Lung Cancer Reports.

| Organization | Model | Accuracy | Precision | Sensitivity | Specificity | F1 Score |
|---|---|---|---|---|---|---|
| | | | | pN | | |
| Others (Alibaba Group & 01.AI) | yi:34b | 0.87 | 0.91 | 0.87 | 0.96 | 0.89 |
| | qwen2:72b-instruct | 0.89 | 0.92 | 0.89 | 0.97 | 0.89 |
| Google DeepMind | gemma:7b-instruct | 0.70 | 0.86 | 0.70 | 0.92 | 0.76 |
| | gemma:2b-instruct | 0.68 | 0.73 | 0.68 | 0.89 | 0.70 |
| Meta | llama3:70b-instruct | 0.93 | 0.94 | 0.93 | 0.98 | 0.93 |
| | llama3.1:70b | 0.89 | 0.94 | 0.89 | 0.97 | 0.91 |
| Mistral AI | mixtral:instruct | 0.85 | 0.92 | 0.85 | 0.96 | 0.88 |
| | mistral-openorca:latest | 0.73 | 0.90 | 0.73 | 0.93 | 0.79 |
| OpenAI | gpt-4o-mini | 0.94 | 0.93 | 0.94 | 0.98 | 0.93 |
| | gpt-4o-2024-05-13 | 0.93 | 0.94 | 0.93 | 0.98 | 0.94 |
| | gpt-3.5-turbo-0125 | 0.91 | 0.92 | 0.91 | 0.98 | 0.91 |

Table S7: Performance of LLMs in Extracting Tumor Staging Information from Lung Cancer Reports.

| Organization | Model | Tumor Stage | | | | |
| | | Accuracy | Precision | Sensitivity | Specificity | F1 Score |
|---|---|---|---|---|---|---|
| Others (Alibaba Group & 01.AI) | yi:34b | 0.74 | 0.74 | 0.74 | 0.90 | 0.74 |
| | qwen2:72b-instruct | 0.80 | 0.81 | 0.80 | 0.93 | 0.80 |
| Google DeepMind | gemma:7b-instruct | 0.56 | 0.55 | 0.56 | 0.80 | 0.51 |
| | gemma:2b-instruct | 0.48 | 0.46 | 0.48 | 0.76 | 0.37 |
| Meta | llama3:70b-instruct | 0.83 | 0.83 | 0.83 | 0.93 | 0.83 |
| | llama3.1:70b | 0.80 | 0.84 | 0.80 | 0.93 | 0.82 |
| Mistral AI | mixtral:instruct | 0.78 | 0.79 | 0.78 | 0.91 | 0.78 |
| | mistral-openorca:latest | 0.56 | 0.63 | 0.56 | 0.84 | 0.59 |
| OpenAI | gpt-4o-2024-05-13 | 0.86 | 0.88 | 0.86 | 0.95 | 0.87 |
| | gpt-4o-mini | 0.86 | 0.87 | 0.86 | 0.95 | 0.86 |
| | gpt-3.5-turbo-0125 | 0.82 | 0.83 | 0.82 | 0.93 | 0.82 |

Table S8: Performance of LLMs in Extracting Histologic Diagnosis from Lung Cancer Reports.

| Organization | Model | Histologic Diagnosis | | | | |
| | | Accuracy | Precision | Sensitivity | Specificity | F1 Score |
|---|---|---|---|---|---|---|
| Others (Alibaba Group & 01.AI) | yi:34b | 0.96 | 0.97 | 0.96 | 0.99 | 0.96 |
| | qwen2:72b-instruct | 0.97 | 0.97 | 0.97 | 0.99 | 0.97 |
| Google DeepMind | gemma:2b-instruct | 0.57 | 0.75 | 0.57 | 0.81 | 0.53 |
| | gemma:7b-instruct | 0.82 | 0.85 | 0.82 | 0.91 | 0.81 |
| Meta | llama3:70b-instruct | 0.98 | 0.98 | 0.98 | 0.99 | 0.98 |
| | llama3.1:70b | 0.99 | 0.99 | 0.99 | 1.00 | 0.99 |
| Mistral AI | mixtral:instruct | 0.92 | 0.96 | 0.92 | 0.98 | 0.94 |
| | mistral-openorca:latest | 0.87 | 0.95 | 0.87 | 0.96 | 0.91 |
| OpenAI | gpt-4o-mini | 0.98 | 0.98 | 0.98 | 0.99 | 0.98 |
| | gpt-3.5-turbo-0125 | 0.98 | 0.98 | 0.98 | 0.99 | 0.98 |
| | gpt-4o-2024-05-13 | 0.98 | 0.98 | 0.98 | 0.99 | 0.98 |

# Benchmarking Results for ECG Information Extraction

Table S9: Overall Performance of LLMs on ECG dataset: Information Extraction Across All Tasks

| Organization | Model | Overall Performance | | | | | |
| | | Accuracy | Precision | Sensitivity | Specificity | F1 Score | Running Time (min) |
|---|---|---|---|---|---|---|---|
| Others (Alibaba Group & 01.AI) | yi:34b | 0.92 | 0.74 | 0.69 | 0.97 | 0.71 | 9.20 |
| | qwen2:72b-instruct | 0.92 | 0.96 | 0.57 | 1.00 | 0.60 | 9.60 |
| Google DeepMind | gemma:7b-instruct | 0.73 | 0.51 | 0.52 | 0.78 | 0.49 | 577.70 |
| | gemma:2b-instruct | 0.88 | 0.61 | 0.60 | 0.94 | 0.60 | 7.00 |
| Meta | llama3:70b-instruct | 0.92 | 0.96 | 0.57 | 1.00 | 0.60 | 9.40 |
| | llama3.1:70b | 0.95 | 0.96 | 0.72 | 1.00 | 0.79 | 9.30 |
| Mistral AI | mixtral:instruct | 0.73 | 0.54 | 0.59 | 0.76 | 0.53 | 20.40 |
| | mistral-openorca:latest | 0.40 | 0.53 | 0.60 | 0.36 | 0.36 | 2.10 |
| OpenAI | gpt-4o-mini | 0.94 | 0.97 | 0.68 | 1.00 | 0.75 | 4.30 |
| | gpt-4o-2024-05-13 | 0.92 | 0.96 | 0.55 | 1.00 | 0.57 | 6.00 |
| | gpt-3.5-turbo-1106 | 0.95 | 0.97 | 0.71 | 1.00 | 0.79 | 5.90 |

Table S10: Performance of LLMs in Extracting Atrial Fibrillation (AF) Information from ECG Reports.

| Organization | Model | Atrial Fibrillation | | | | |
| --- | --- | --- | --- | --- | --- | --- |
| | | Accuracy | Precision | Sensitivity | Specificity | F1 Score |
| Others (Alibaba Group & 01.AI) | yi:34b-8K | 0.92 | 0.94 | 0.97 | 0.97 | 0.95 |
| | qwen2:72b-instruct-8K | 0.92 | 0.92 | 1.00 | 1.00 | 0.96 |
| Google DeepMind | gemma:7b-instruct-8K | 0.73 | 0.92 | 0.78 | 0.78 | 0.84 |
| | gemma:2b-instruct-8K | 0.88 | 0.93 | 0.94 | 0.94 | 0.93 |
| Meta | llama3:70b-instruct-8K | 0.92 | 0.92 | 1.00 | 1.00 | 0.96 |
| | llama3.1:70b-8K | 0.95 | 0.95 | 1.00 | 1.00 | 0.97 |
| Mistral AI | mixtral:instruct-8K | 0.73 | 0.93 | 0.76 | 0.76 | 0.84 |
| | mistral-openorca:latest-8K | 0.40 | 0.96 | 0.36 | 0.36 | 0.52 |
| OpenAI | gpt-4o-mini | 0.94 | 0.94 | 1.00 | 1.00 | 0.97 |
| | gpt-4o-2024-05-13 | 0.92 | 0.92 | 1.00 | 1.00 | 0.96 |
| | gpt-3.5-turbo-1106 | 0.95 | 0.95 | 1.00 | 1.00 | 0.97 |

Table S11: Performance of LLMs in Extracting Non-Atrial Fibrillation (Non-Afib) Information from ECG Reports

| Organization | Model | Non-Atrial Fibrillation | | | | |
| --- | --- | --- | --- | --- | --- | --- |
| | | Accuracy | Precision | Sensitivity | Specificity | F1 Score |
| Others (Alibaba Group & 01.AI) | yi:34b-8K | 0.92 | 0.53 | 0.41 | 0.97 | 0.96 |
| | qwen2:72b-instruct-8K | 0.92 | 1.00 | 0.13 | 1.00 | 0.23 |
| Google DeepMind | gemma:7b-instruct-8K | 0.73 | 0.10 | 0.26 | 0.78 | 0.15 |
| | gemma:2b-instruct-8K | 0.88 | 0.29 | 0.26 | 0.94 | 0.27 |
| Meta | llama3:70b-instruct-8K | 0.92 | 1.00 | 0.13 | 1.00 | 0.23 |
| | llama3.1:70b-8K | 0.95 | 0.96 | 0.44 | 1.00 | 0.61 |
| Mistral AI | mixtral:instruct-8K | 0.73 | 0.15 | 0.43 | 0.76 | 0.22 |
| | mistral-openorca:latest-8K | 0.40 | 0.10 | 0.83 | 0.35 | 0.19 |
| OpenAI | gpt-4o-mini | 0.94 | 1.00 | 0.36 | 1.00 | 0.53 |
| | gpt-4o-2024-05-13 | 0.92 | 1.00 | 0.10 | 1.00 | 0.18 |
| | gpt-3.5-turbo-1106 | 0.95 | 1.00 | 0.43 | 1.00 | 0.60 |

# Benchmarking Results for SDOH Information Extraction

### Employment

Table S12: Overall Performance of LLMs on the SDOH dataset (Employment): Information Extraction Across All Tasks

| Organization | Model | Overall Performance | | | | | |
| --- | --- | --- | --- | --- | --- | --- | --- |
| | | Accuracy | Precision | Sensitivity | Specificity | F1 Score | Running Time (min) |
| Others (Alibaba Group & 01.AI) | yi:34b | 0.88 | 0.76 | 0.81 | 0.91 | 0.76 | 24.18 |
| | qwen2:72b-instruct | 0.90 | 0.79 | 0.86 | 0.93 | 0.81 | 30.90 |
| Google DeepMind | gemma:7b-instruct | 0.84 | 0.71 | 0.69 | 0.85 | 0.69 | 117.08 |
| | gemma:2b-instruct | 0.53 | 0.67 | 0.42 | 0.68 | 0.28 | 7.02 |
| Meta | llama3:70b-instruct | 0.90 | 0.78 | 0.87 | 0.93 | 0.81 | 28.04 |
| | llama3.1:70b | 0.91 | 0.80 | 0.87 | 0.93 | 0.82 | 27.66 |
| Mistral AI | mixtral:instruct | 0.87 | 0.74 | 0.81 | 0.91 | 0.75 | 40.73 |
| | mistral-openorca:latest | 0.80 | 0.72 | 0.75 | 0.88 | 0.66 | 8.11 |
| OpenAI | GPT-4o-mini | 0.91 | 0.79 | 0.87 | 0.94 | 0.82 | 12.59 |
| | gpt-4o-2024-05-16 | 0.91 | 0.79 | 0.86 | 0.93 | 0.82 | 19.59 |
| | gpt-3.5-turbo-1109 | 0.91 | 0.79 | 0.87 | 0.93 | 0.82 | 19.55 |

Table S13: Performance of LLMs in Extracting Employment Status (Employed) from SDOH Reports.

| Organization | Model | Accuracy | Precision | Employed Sensitivity | Specificity | F1 Score |
|---|---|---|---|---|---|---|
| Others (Alibaba Group & 01.AI) | yi:34b | 0.88 | 0.49 | 0.88 | 0.88 | 0.63 |
| | qwen2:72b-instruct | 0.91 | 0.58 | 0.89 | 0.92 | 0.70 |
| Google DeepMind | gemma:7b-instruct | 0.89 | 0.51 | 0.64 | 0.92 | 0.57 |
| | gemma:2b-instruct | 0.91 | 0.74 | 0.27 | 0.99 | 0.39 |
| Meta | llama3:70b-instruct | 0.91 | 0.55 | 0.89 | 0.91 | 0.68 |
| | llama3.1:70b | 0.92 | 0.61 | 0.89 | 0.93 | 0.72 |
| Mistral AI | mixtral:instruct | 0.86 | 0.43 | 0.85 | 0.86 | 0.58 |
| | mistral-openorca:latest | 0.73 | 0.29 | 0.94 | 0.71 | 0.45 |
| OpenAI | gpt-4o-mini | 0.91 | 0.58 | 0.89 | 0.92 | 0.70 |
| | gpt-4o-2024-05-13 | 0.91 | 0.59 | 0.86 | 0.92 | 0.70 |
| | gpt-3.5-turbo-1106 | 0.92 | 0.59 | 0.87 | 0.92 | 0.71 |

Table S14: Performance of LLMs in Extracting Unemployment Status from SDOH Reports.

| Organization | Model | Accuracy | Precision | Unemployed Sensitivity | Specificity | F1 Score |
|---|---|---|---|---|---|---|
| Others (Alibaba Group & 01.AI) | yi:34b | 0.88 | 0.86 | 0.70 | 0.96 | 0.77 |
| | qwen2:72b-instruct | 0.91 | 0.84 | 0.84 | 0.94 | 0.84 |
| Google DeepMind | gemma:7b-instruct | 0.84 | 0.80 | 0.56 | 0.95 | 0.66 |
| | gemma:2b-instruct | 0.30 | 0.28 | 0.98 | 0.05 | 0.43 |
| Meta | llama3:70b-instruct | 0.91 | 0.83 | 0.87 | 0.93 | 0.85 |
| | llama3.1:70b | 0.91 | 0.84 | 0.84 | 0.94 | 0.84 |
| Mistral AI | mixtral:instruct | 0.91 | 0.87 | 0.78 | 0.95 | 0.82 |
| | mistral-openorca:latest | 0.87 | 0.90 | 0.57 | 0.98 | 0.70 |
| OpenAI | gpt-4o-mini | 0.92 | 0.82 | 0.89 | 0.93 | 0.86 |
| | gpt-4o-2024-05-13 | 0.92 | 0.84 | 0.87 | 0.94 | 0.85 |
| | gpt-3.5-turbo-1106 | 0.92 | 0.83 | 0.88 | 0.93 | 0.85 |

Table S15: Performance of LLMs in Extracting Unknown Employment Status from SDOH Reports.

| Organization | Model | Accuracy | Precision | Unknown Sensitivity | Specificity | F1 Score |
|---|---|---|---|---|---|---|
| Others (Alibaba Group & 01.AI) | yi:34b | 0.87 | 0.92 | 0.86 | 0.88 | 0.89 |
| | qwen2:72b-instruct | 0.88 | 0.95 | 0.85 | 0.92 | 0.90 |
| Google DeepMind | gemma:7b-instruct | 0.80 | 0.81 | 0.88 | 0.67 | 0.84 |
| | gemma:2b-instruct | 0.39 | 1.00 | 0.00 | 1.00 | 0.00 |
| Meta | llama3:70b-instruct | 0.89 | 0.97 | 0.84 | 0.97 | 0.90 |
| | llama3.1:70b | 0.88 | 0.94 | 0.86 | 0.91 | 0.90 |
| Mistral AI | mixtral:instruct | 0.85 | 0.93 | 0.81 | 0.91 | 0.87 |
| | mistral-openorca:latest | 0.81 | 0.96 | 0.72 | 0.96 | 0.82 |
| OpenAI | gpt-4o-mini | 0.89 | 0.97 | 0.84 | 0.96 | 0.90 |
| | gpt-4o-2024-05-13 | 0.89 | 0.95 | 0.85 | 0.94 | 0.90 |
| | gpt-3.5-turbo-1106 | 0.89 | 0.96 | 0.85 | 0.95 | 0.90 |

**Housing**

Table S16: Overall Performance of LLMs on the SDOH dataset (Housing): Information Extraction Across All Tasks

| Organization | Model | Accuracy | Precision | Sensitivity | Specificity | F1 Score | Running Time (min) |
|---|---|---|---|---|---|---|---|
| | | | | Overal Performance | | | |
| Others (Alibaba Group & 01.AI) | yi:34b | 0.89 | 0.64 | 0.85 | 0.89 | 0.66 | 24.18 |
| | qwen2:72b-instruct | 0.92 | 0.78 | 0.88 | 0.91 | 0.82 | 30.90 |
| Google DeepMind | gemma:7b-instruct | 0.77 | 0.51 | 0.73 | 0.80 | 0.53 | 117.08 |
| | gemma:2b-instruct | 0.48 | 0.49 | 0.45 | 0.70 | 0.17 | 7.02 |
| Meta | llama3:70b-instruct | 0.93 | 0.73 | 0.89 | 0.92 | 0.76 | 28.04 |
| | llama3.1:70b | 0.92 | 0.71 | 0.89 | 0.92 | 0.75 | 27.66 |
| Mistral AI | mixtral:instruct | 0.87 | 0.74 | 0.83 | 0.86 | 0.78 | 40.73 |
| | mistral-openorca:latest | 0.85 | 0.66 | 0.78 | 0.82 | 0.65 | 8.11 |
| OpenAI | GPT-4o-mini | 0.93 | 0.72 | 0.89 | 0.92 | 0.76 | 12.59 |
| | gpt-4o-2024-05-16 | 0.87 | 0.85 | 0.82 | 0.84 | 0.82 | 19.59 |
| | gpt-3.5-turbo-1109 | 0.87 | 0.84 | 0.81 | 0.84 | 0.81 | 19.55 |

Table S17: Performance of LLMs in Identifying Housing Stability (Having a Place to Live) from SDOH Reports.

| Organization | Model | Accuracy | Precision | Sensitivity | Specificity | F1 Score |
|---|---|---|---|---|---|---|
| | | | | Housing | | |
| Others (Alibaba Group & 01.AI) | yi:34b | 0.84 | 0.87 | 0.87 | 0.78 | 0.87 |
| | qwen2:72b-instruct | 0.89 | 0.88 | 0.95 | 0.78 | 0.91 |
| Google DeepMind | gemma:7b-instruct | 0.68 | 0.81 | 0.64 | 0.76 | 0.71 |
| | gemma:2b-instruct | 0.51 | 0.75 | 0.32 | 0.82 | 0.45 |
| Meta | llama3:70b-instruct | 0.90 | 0.88 | 0.97 | 0.78 | 0.92 |
| | llama3.1:70b | 0.89 | 0.89 | 0.94 | 0.80 | 0.91 |
| Mistral AI | mixtral:instruct | 0.80 | 0.85 | 0.83 | 0.75 | 0.84 |
| | mistral-openorca:latest | 0.78 | 0.76 | 0.95 | 0.50 | 0.84 |
| OpenAI | gpt-4o-mini | 0.90 | 0.90 | 0.94 | 0.83 | 0.92 |
| | gpt-4o-2024-05-13 | 0.81 | 0.79 | 0.96 | 0.57 | 0.87 |
| | gpt-3.5-turbo-1106 | 0.81 | 0.79 | 0.95 | 0.57 | 0.86 |

Table S18: Performance of LLMs in Extracting Homelessness Status (Lacking a Stable Place to Live) from SDOH Reports.

| Organization | Model | Accuracy | Precision | Sensitivity | Specificity | F1 Score |
|---|---|---|---|---|---|---|
| | | | | Homeless | | |
| Others (Alibaba Group & 01.AI) | yi:34b | 0.96 | 0.18 | 0.93 | 0.96 | 0.30 |
| | qwen2:72b-instruct | 0.99 | 0.57 | 0.93 | 0.99 | 0.70 |
| Google DeepMind | gemma:7b-instruct | 0.95 | 0.15 | 0.86 | 0.95 | 0.26 |
| | gemma:2b-instruct | 0.29 | 0.01 | 1.00 | 0.28 | 0.03 |
| Meta | llama3:70b-instruct | 0.98 | 0.35 | 0.93 | 0.98 | 0.51 |
| | llama3.1:70b | 0.98 | 0.32 | 0.93 | 0.98 | 0.47 |
| Mistral AI | mixtral:instruct | 0.99 | 0.65 | 0.93 | 0.99 | 0.76 |
| | mistral-openorca:latest | 0.98 | 0.33 | 0.93 | 0.98 | 0.49 |
| OpenAI | gpt-4o-mini | 0.98 | 0.34 | 0.93 | 0.98 | 0.50 |
| | gpt-4o-2024-05-13 | 1.00 | 0.87 | 0.93 | 1.00 | 0.90 |
| | gpt-3.5-turbo-1106 | 1.00 | 0.87 | 0.93 | 1.00 | 0.90 |

Table S19: Performance of LLMs in Extracting Unknown Housing Status from SDOH Reports.

| Organization | Model | Accuracy | Precision | Sensitivity | Specificity | F1 Score |
|---|---|---|---|---|---|---|
| | | | | Unknown | | |
| Others (Alibaba Group & 01.AI) | yi:34b | 0.87 | 0.86 | 0.75 | 0.90 | 0.80 |
| | qwen2:72b-instruct | 0.89 | 0.91 | 0.77 | 0.96 | 0.83 |
| Google DeepMind | gemma:7b-instruct | 0.68 | 0.55 | 0.69 | 0.68 | 0.61 |
| | gemma:2b-instruct | 0.64 | 0.69 | 0.02 | 0.99 | 0.03 |
| Meta | llama3:70b-instruct | 0.91 | 0.97 | 0.76 | 0.99 | 0.85 |
| | llama3.1:70b | 0.90 | 0.94 | 0.79 | 0.97 | 0.86 |
| Mistral AI | mixtral:instruct | 0.80 | 0.73 | 0.73 | 0.91 | 0.73 |
| | mistral-openorca:latest | 0.78 | 0.90 | 0.46 | 0.85 | 0.61 |
| OpenAI | gpt-4o-mini | 0.91 | 0.92 | 0.81 | 0.93 | 0.86 |
| | gpt-4o-2024-05-13 | 0.81 | 0.88 | 0.56 | 0.96 | 0.69 |
| | gpt-3.5-turbo-1106 | 0.81 | 0.87 | 0.56 | 0.95 | 0.68 |

## A.8 Confusion Matrices

Confusion Matrices for Tumor Stage

Figure S1: Confusion matrices for 11 LLMs on the Lung Cancer Overall Tumor Stage extraction task from TCGA pathology reports. Each subplot details an LLM's performance, with rows representing true AJCC 7th edition stage categories and columns representing predicted stage categories. Cell values indicate instance counts; darker shades correspond to higher counts. These matrices reveal model accuracy and misclassification patterns for overall tumor staging.

Confusion Matrices for Pt

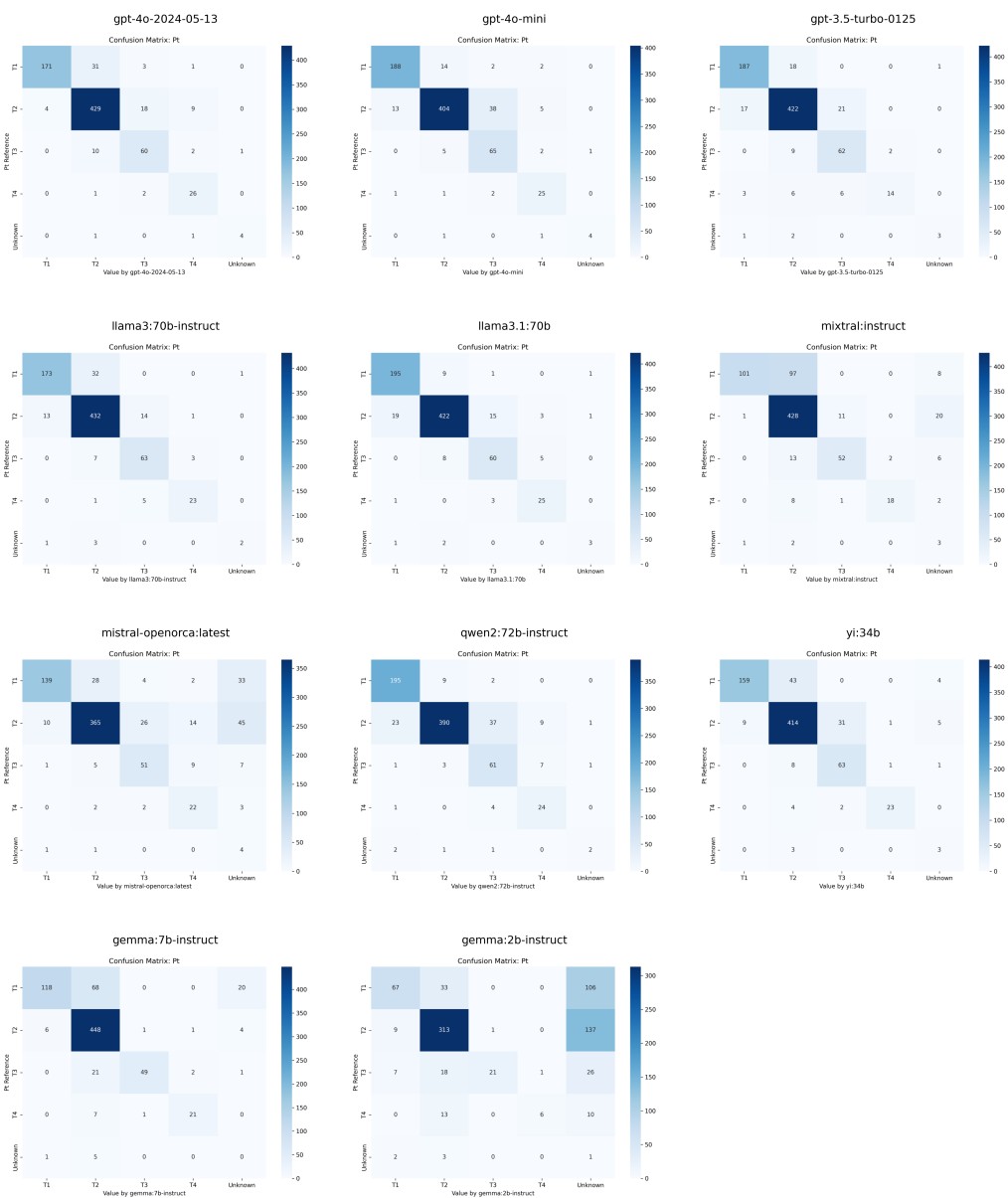

Figure S2: Confusion matrices for 11 LLMs on the Lung Cancer primary tumor (pT) classification task from TCGA pathology reports. Each subplot details an LLM's performance, with rows representing true AJCC 7th edition pT categories (e.g., pT1-pT4) and columns representing predicted categories. Cell values indicate instance counts; darker shades correspond to higher counts. These matrices highlight model accuracy and error types in pT classification.

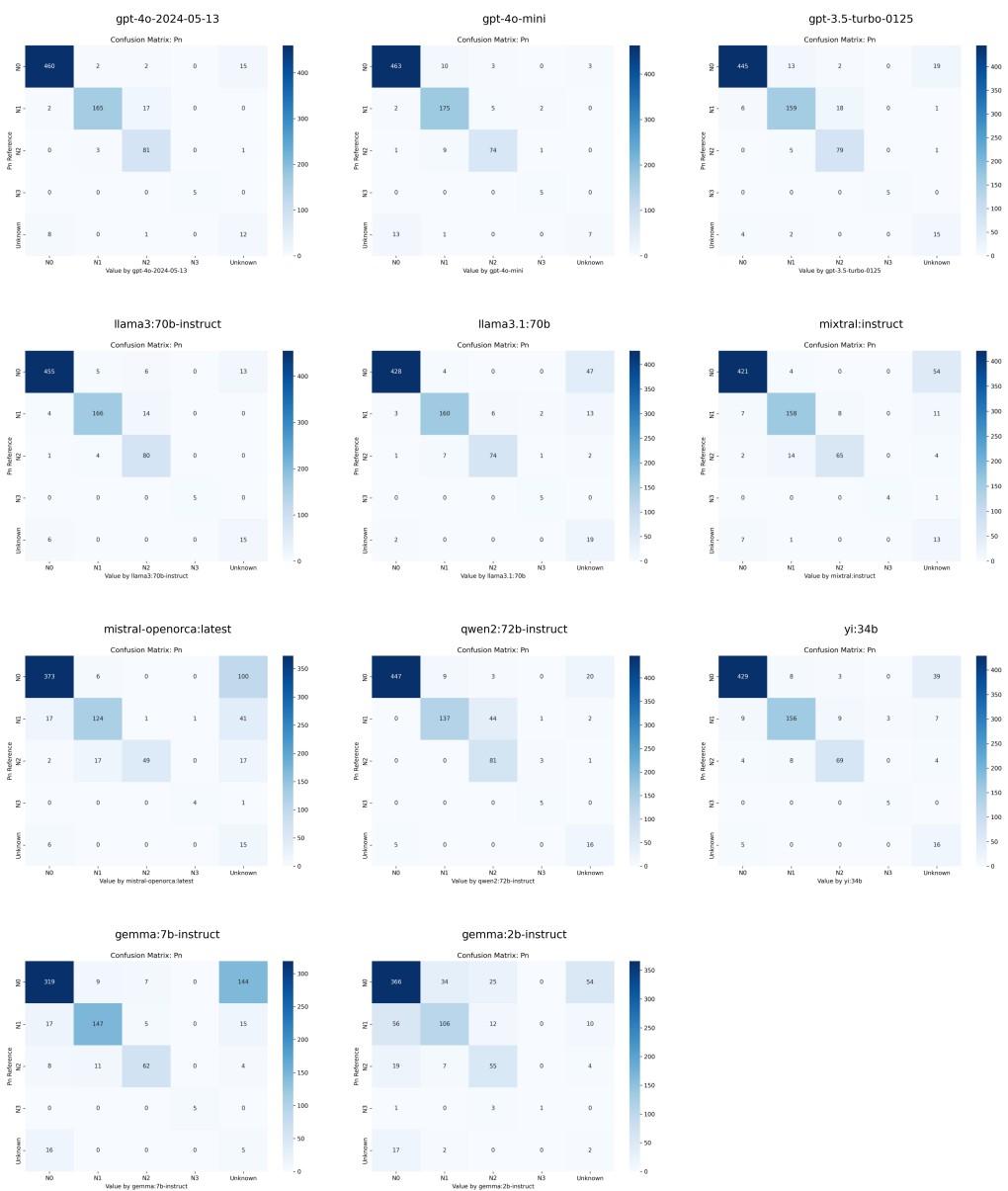

Figure S3: Confusion matrices for 11 LLMs on the Lung Cancer regional lymph node (pN) classification task from TCGA pathology reports. Each subplot details an LLM's performance, with rows representing true AJCC 7th edition pN categories (e.g., pN0-pN3) and columns representing predicted categories. Cell values indicate instance counts; darker shades correspond to higher counts. These matrices illustrate model accuracy and misclassification patterns for pN classification.

Confusion Matrices for Histologic Diagnosis

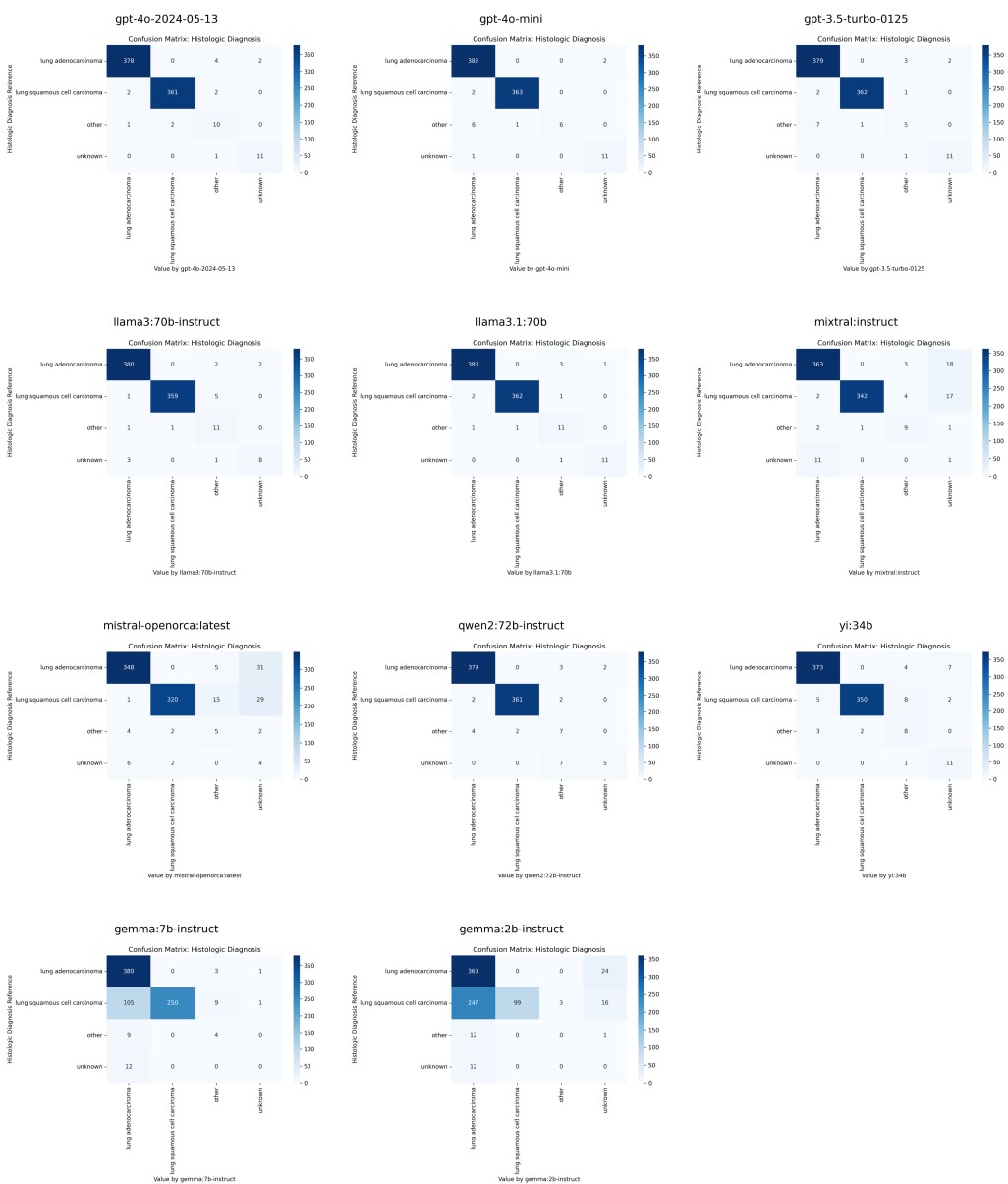

Figure S4: Confusion matrices for 11 LLMs on the Lung Cancer Histologic Diagnosis extraction task (e.g., Adenocarcinoma, Squamous Cell Carcinoma) from TCGA pathology reports. Each subplot details an LLM's performance, with rows representing true histologic types and columns representing predicted types. Cell values indicate instance counts; darker shades correspond to higher counts. These matrices show model accuracy and error patterns in identifying lung cancer histologies.

Confusion Matrices for Diagnosis

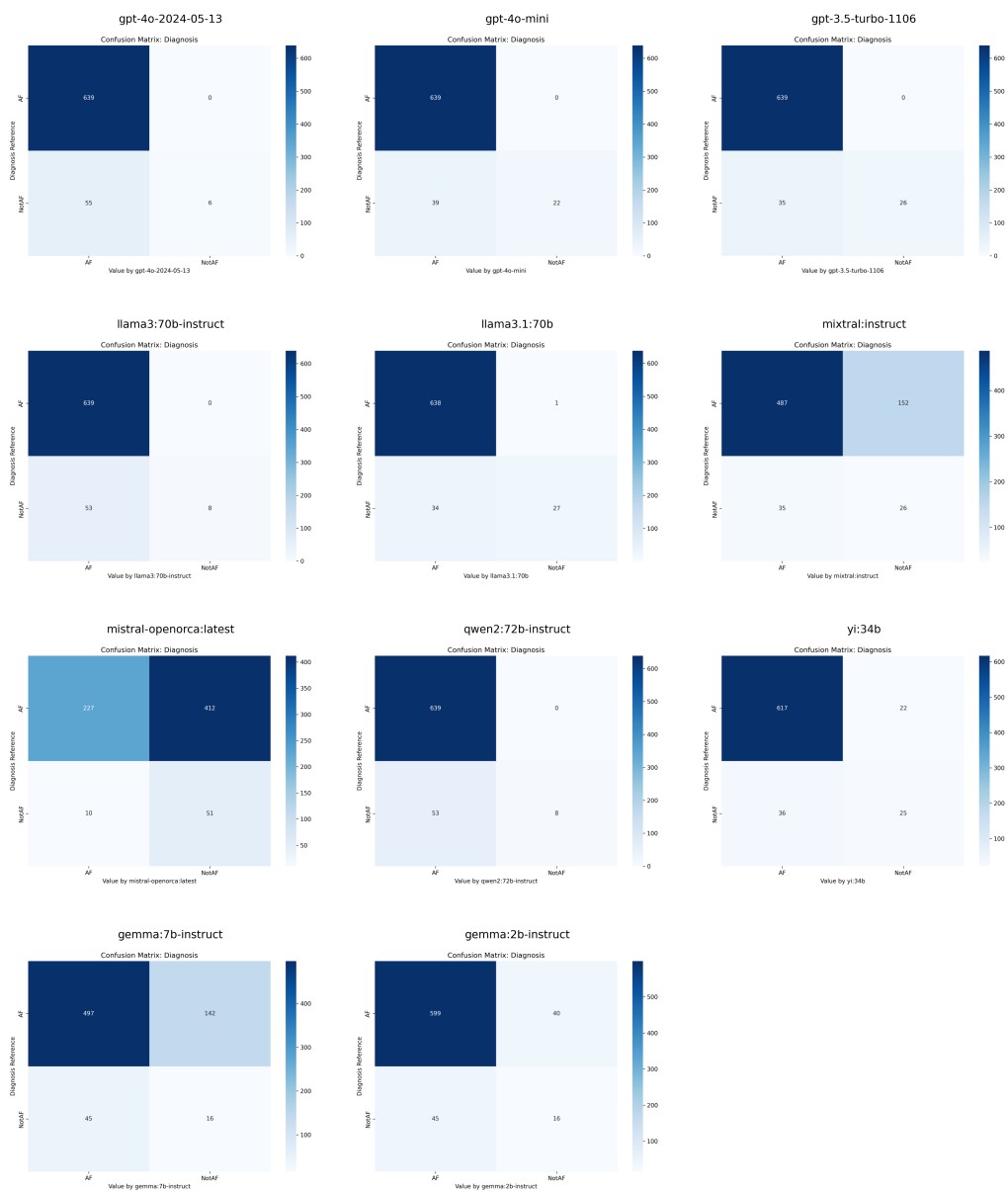

Figure S5: Confusion matrices for 11 LLMs on the Atrial Fibrillation (AF) detection task from MIMIC-IV-ECG reports. Each subplot displays the performance of an individual LLM, with rows representing true labels (AF, Non-AF) and columns representing predicted labels. Cell values indicate instance counts, and darker shades correspond to higher counts. These matrices provide a detailed view of classification accuracy, including true positives (correct AF detection), true negatives (correct Non-AF identification), false positives (Non-AF incorrectly identified as AF), and false negatives (AF incorrectly identified as Non-AF).

# Confusion Matrices for Employment

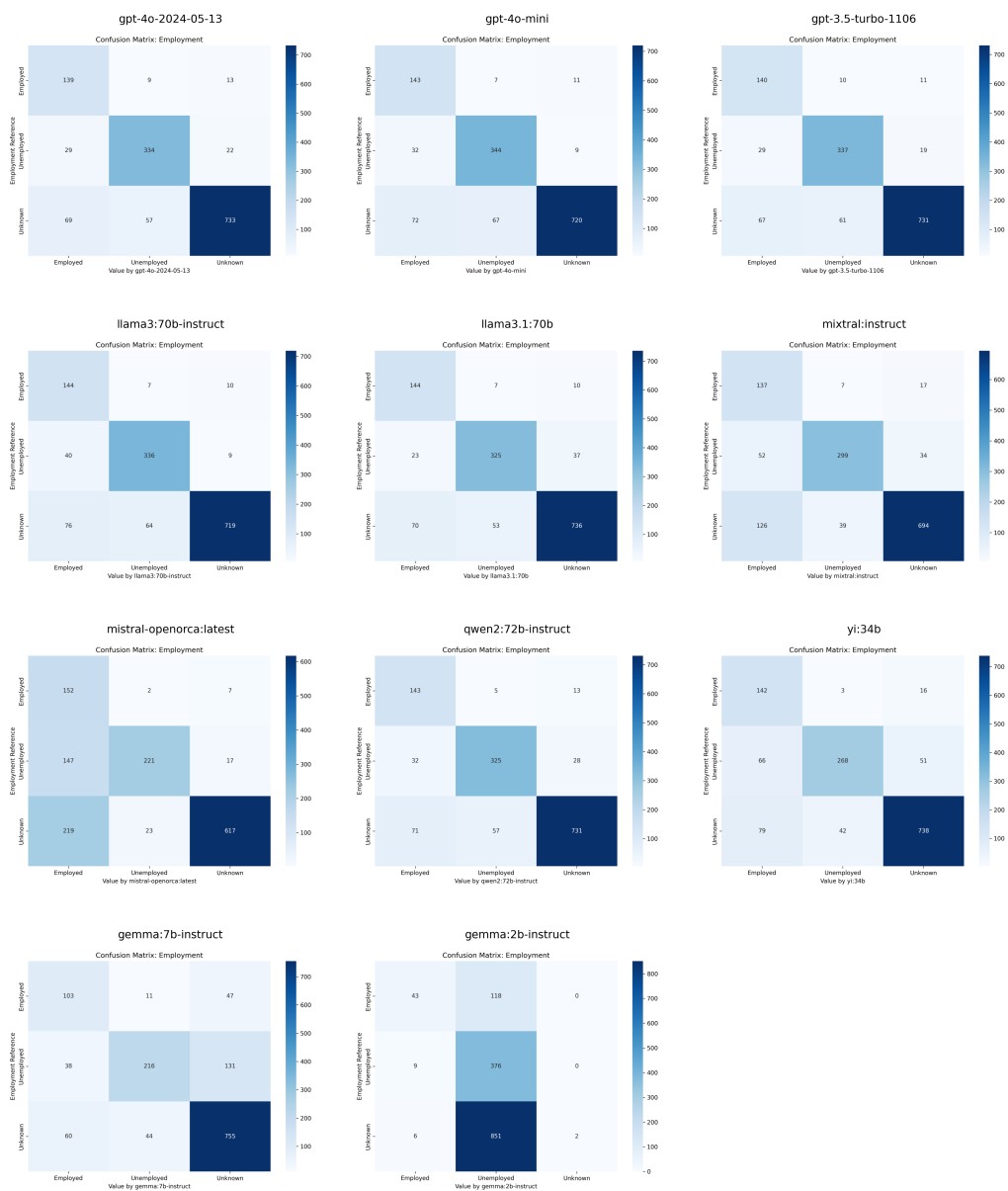

Figure S6: Confusion matrices for 11 LLMs on the Social Determinants of Health (SDOH) Employment Status extraction task from MIMIC clinical notes. Each subplot details an LLM's performance, with rows representing true labels (e.g., "Employed", "Unemployed", "Unknown") and columns representing predicted labels. Cell values indicate instance counts; darker shades correspond to higher counts. These matrices illustrate model-specific classification accuracy and common error patterns for employment status identification.

# Confusion Matrices for Housing

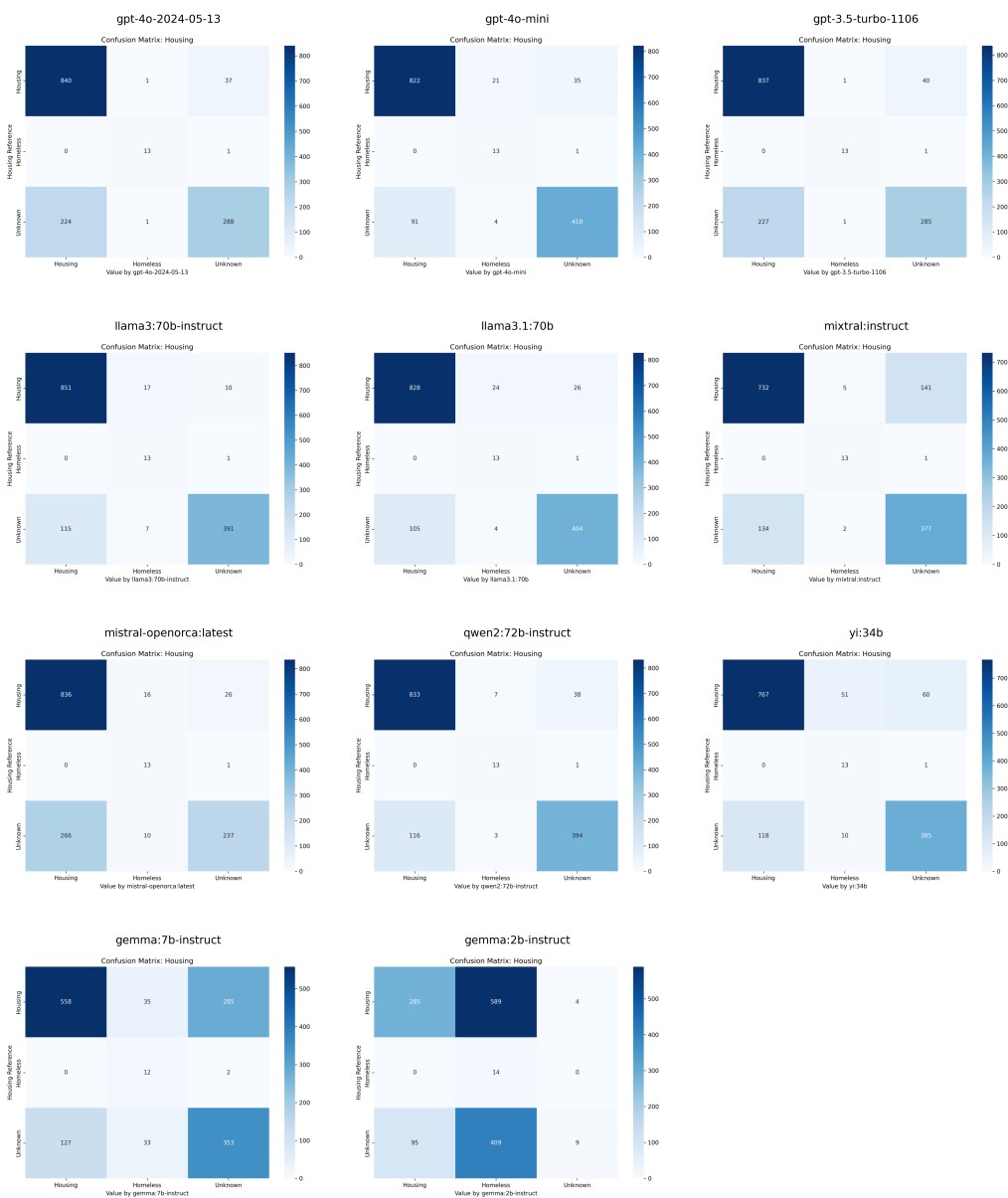

Figure S7: Confusion matrices for 11 LLMs on the Social Determinants of Health (SDOH) Housing Status extraction task from MIMIC clinical notes. Each subplot details an LLM's performance, with rows representing true labels (e.g., "Housing", "Homeless", "Unknown") and columns representing predicted labels. Cell values indicate instance counts; darker shades correspond to higher counts. These matrices illustrate model-specific classification accuracy and common error patterns for housing status identification.

