# OpenReview forum: "ClinBench: A Standardized Multi-Domain Framework for Evaluating Large Language Models in Clinical Information Extraction"
_NeurIPS.cc/2025/Datasets_and_Benchmarks_Track — NeurIPS 2025 Datasets and Benchmarks Track poster_

### Official Review · Reviewer_PoFR · 2025-06-06

**Rating:** 5
**Confidence:** 3

**Summary:**

The paper proposes a benchmarking framework for evaluating LLMs on structured information extraction tasks from unstructured texts. The framework attempts to standardize the evaluation pipeline by enforcing consistent formats of data input, LLM prompts and LLM output. Using the standardized evaluation they evaluate 11 popular LLMs for three clinical extraction tasks and find significant performance-efficiency trade-offs.

**Dataset Code Accessibility:**

Yes

**Dataset Code Comments:**

the github repo provided has clear instructions on how to access the data and use the code to evaluate LLMs

Although I did not go through the whole process of running the evaluation myself because it is quite involved process to get the benchmark datasets themselves.

**Ethical Considerations:**

No, there are no or only very minor ethics concerns

**Final Justification:**

After going through all the discussion, I feel like the authors have sufficiently addressed the concerns raised by all the reviewers.
I am keeping my initial score of Accept (5).

**Limitations Weaknesses:**

Major weakness: Runtime comparison of open source LLMs with proprietary LLMs is not fair. LLM inference times depend on how much engineering goes into hosting the LLM, and may not depend on the LLM itself. A more optimized hosting service (like https://github.com/vllm-project/vllm) might make inference way faster on open source LLMs.

Minor weakness: Proposed standardized pipeline would need wide scale adoption to have high impact: because different medical text datasets have restrictions, using the proposed evaluation framework would have to be adopted by a wide range of medical research institutes to enable different benchmarking datasets to be onboarded on ClinBench. Although extensive instructions are given in the code documentation, in its current form, it is not super straightforward to evaluate my model on tasks using ClinBench.

**Strengths Contributions:**

1. Propose a standardized evaluation framework: if widely adopted and extended to many publicly available clinical text extraction datasets, could serve as an important benchmark for the community to evaluate their models.
- standardized data input, prompting and output formats for different tasks can help ClinBench extend to other tasks.
- could help researchers to evaluate their models on the available tasks on ClinBench

2. Comprehensive analysis of different LLMs (including both open source and proprietary)/

3, Well written paper: The paper is well organized and easy to understand with clear figures, tables and analysis.

4. Well documented open source evaluation code of framework (I don't know how strong of a strength is it because it is expected in Datasets and Benchmarks Track anyway)

---

> ### Author Rebuttal · Authors · 2025-07-30
>
> **`Reviewer`**: *The paper proposes a benchmarking framework for evaluating LLMs on structured information extraction tasks from unstructured texts. The framework attempts to standardize the evaluation pipeline by enforcing consistent formats of data input, LLM prompts and LLM output. Using the standardized evaluation they evaluate 11 popular LLMs for three clinical extraction tasks and find significant performance-efficiency trade-offs.*
>
> ### *Strengths Contributions:*
>
> - *Propose a standardized evaluation framework: if widely adopted and extended to many publicly available clinical text extraction datasets, could serve as an important benchmark for the community to evaluate their models.*
> - *Standardized data input, prompting and output formats for different tasks can help ClinBench extend to other tasks.*
> - *Could help researchers to evaluate their models on the available tasks on ClinBench.*
> - *Comprehensive analysis of different LLMs (including both open source and proprietary).*
> - *Well written paper: The paper is well organized and easy to understand with clear figures, tables and analysis.*
> - *Well documented open source evaluation code of framework (I don't know how strong of a strength is it because it is expected in Datasets and Benchmarks Track anyway)*
>
> **`Answer`**: We are incredibly grateful for your positive assessment and strong support for our work. We thank you for recognizing our framework as a potentially "important benchmark for the community." Your feedback on the limitations is insightful, and we have addressed these points.
>
> ---
>
> **`Reviewer`**: *Major weakness: Runtime comparison of open source LLMs with proprietary LLMs is not fair. LLM inference times depend on how much engineering goes into hosting the LLM, and may not depend on the LLM itself. A more optimized hosting service (like https://github.com/vllm-project/vllm) might make inference way faster on open source LLMs.*
>
> **`Answer`**: This is an excellent and important point. We completely agree that runtime is hardware-dependent and can be an unfair comparison, a limitation we noted in our original manuscript (Section 12):
> > “Second, reported runtime metrics are inherently hardware-dependent and may vary across different computational environments.”
>
> To address this directly and provide a fairer, more hardware-agnostic metric, we have now performed a comprehensive token cost analysis for all OpenAI models.
>
> | Dataset | Model | Prompt Tokens | Completion Tokens | Total Tokens | Prompt Cost ($) | Completion Cost ($) | Total Estimated Cost ($) |
> |---------|--------|----------------|--------------------|---------------|------------------|----------------------|---------------------------|
> | SDOH    | gpt-3.5-turbo-1106   | 753207  | 33044   | 786251   | 0.75  | 0.07  | 0.82  |
> | SDOH    | gpt-4o-2024-05-13    | 753207  | 28506   | 781713   | 3.77  | 0.43  | 4.19  |
> | SDOH    | gpt-4o-mini          | 753207  | 32932   | 786139   | 0.11  | 0.02  | 0.13  |
> | ECG     | gpt-3.5-turbo-1106   | 631568  | 3869    | 635437   | 0.63  | 0.01  | 0.64  |
> | ECG     | gpt-4o-2024-05-13    | 631568  | 3855    | 635423   | 3.16  | 0.06  | 3.22  |
> | ECG     | gpt-4o-mini          | 631568  | 13322   | 644890   | 0.09  | 0.01  | 0.10  |
> | Lung    | gpt-3.5-turbo-1106   | 1747615 | 61349   | 1808964  | 1.75  | 0.12  | 1.87  |
> | Lung    | gpt-4o-2024-05-13    | 1747615 | 61656   | 1809271  | 8.74  | 0.92  | 9.66  |
> | Lung    | gpt-4o-mini          | 1747615 | 61539   | 1809154  | 0.26  | 0.04  | 0.30  |
>
> ---
>
> **`Reviewer`**: *Minor weakness: Proposed standardized pipeline would need wide scale adoption to have high impact: because different medical text datasets have restrictions, using the proposed evaluation framework would have to be adopted by a wide range of medical research institutes to enable different benchmarking datasets to be onboarded on ClinBench. Although extensive instructions are given in the code documentation, in its current form, it is not super straightforward to evaluate my model on tasks using ClinBench.*
>
> **`Answer`**: We thank you for this thoughtful comment on the practical challenges of adoption. We acknowledge that this is a key challenge for any new benchmark. Our goal with ClinBench was to provide a solid, well-documented first step. By open-sourcing the code and using a modular, YAML-based configuration, we aimed to lower the barrier to entry as much as possible, and we are committed to improving usability based on community feedback.
>
> ---
>
> `Thank you again for your valuable feedback and your strong support.`

---

> > ### Comment · Reviewer_PoFR · 2025-08-04
> >
> > I have gone through all the reviews and author responses. I think that the authors have sufficiently addressed the points raised by the reviewers. I am inclined to keep my score.

---

> > > ### Author Response · Authors · 2025-08-07
> > >
> > > Thank you for your thoughtful review and for taking the time to consider our responses. We appreciate your assessment that the concerns raised by the reviewers have been sufficiently addressed, and we are grateful for your decision to maintain a score of 5 (Accept). Your feedback and support are sincerely appreciated.

---

### Official Review · Reviewer_ea25 · 2025-07-01

**Rating:** 4
**Confidence:** 5

**Summary:**

ClinBench proposes a standardized, open-source framework for evaluating LLMs in clinical information extraction across multiple domains (oncology, cardiology, SDOH). It addresses reproducibility gaps in clinical NLP by:

- Standardizing inputs (structured CSV datasets from public sources: TCGA, MIMIC-IV-ECG, MIMIC-III).

- Dynamic YAML-based prompting for task definition, domain knowledge, and JSON schema-enforced output validation.

- Comprehensive evaluation of 11 LLMs (proprietary and open-source) across four tasks (lung cancer staging, AF detection, SDOH employment/housing extraction), reporting F1, runtime, and performance-efficiency trade-offs.

**Additional Feedback:**

1. Ablation study on YAML-based prompting: To highlight the value of structured prompting, it's better to compare ClinBench’s YAML approach against raw, unstructured prompts.This would empirically demonstrate YAML’s role in reducing prompt drift and improving reliability.
2. Add a case study on terminological challenge. Use misclassification analysis to identify where ambiguity harms performance.

**Dataset Code Accessibility:**

Yes

**Dataset Code Comments:**

Datasets are publicly accessible. The framework of codes is well-structured, modular, and easy to read.

**Ethical Considerations:**

No, there are no or only very minor ethics concerns

**Final Justification:**

As a reviewer with 10+ years in clinical NLP (industry/academia), I recommend borderline acceptance based on these resolved and
unresolved aspects:

1. Resolved Strengths

- Practical Impact: ClinBench’s YAML-driven framework—while not algorithmically novel—provides a simple, standardized, and industrially actionable solution for LLM evaluation in clinical extraction.
- Critical Validation Added: Authors convincingly addressed prior concerns by adding ablation studies (YAML vs. free-text prompts), demonstrating YAML’s  F1 improvement.
- Multi-domain Benchmark: Multi-domain coverage (oncology/cardiology/SDOH), rigorous gold standards (AJCC/MIMIC annotations), and LLM efficiency trade-off analysis fulfill the track’s scope.

2. Unresolved but Track-Appropriate Limitations
- Clinical Safeguards for YAML: While YAML’s schema validation ensures output correctness, clinical auditing of YAML content (e.g., expert validation of AJCC criteria embeddings) remains future work.
- Method Comparisons: No direct RAG/hybrid comparisons were added (industry’s dominant approach), but ClinBench’s focus on standardization (not SOTA pursuit) aligns with the track’s goals.

For the Datasets and Benchmarks Track, ClinBench delivers exceptional value: a turnkey, extensible framework that bridges LLM potential and clinical reliability.

**Limitations Weaknesses:**

Medical concepts often exhibit subtle distinctions. ClinBench does not address how LLMs resolve such ambiguities, nor does it evaluate robustness to synonymy or contextual nuance. This is a critical gap since misclassification could impact clinical decisions.

**Strengths Contributions:**

1. ClinBench handles diverse clinical modalities (pathology reports, ECG interpretations, SDOH notes), demonstrating versatility across domains with distinct terminologies and structures. The framework’s JSON schema validation ensures outputs are structurally consistent and clinically actionable.

2. The YAML-driven agent standardizes task definitions, domain knowledge, and output schemas (e.g., AJCC staging criteria for oncology, AF diagnostic rules for cardiology). This modular approach enhances reproducibility, simplifies benchmarking of new tasks/domains, and mitigates prompt design variability.

3.The evaluation of 11 diverse LLMs provides actionable insights into performance-efficiency trade-offs (e.g., GPT-4o-mini’s optimal speed/accuracy balance vs. LLaMA3.1-70b’s high accuracy at higher computational cost).

---

> ### Author Rebuttal · Authors · 2025-07-30
>
> **`Reviewer`**: *ClinBench proposes a standardized, open-source framework for evaluating LLMs in clinical information extraction across multiple domains (oncology, cardiology, SDOH). It addresses reproducibility gaps in clinical NLP by:*
>
> - *Standardizing inputs (structured CSV datasets from public sources: TCGA, MIMIC-IV-ECG, MIMIC-III).*
> - *Dynamic YAML-based prompting for task definition, domain knowledge, and JSON schema-enforced output validation.*
> - *Comprehensive evaluation of 11 LLMs (proprietary and open-source) across four tasks (lung cancer staging, AF detection, SDOH employment/housing extraction), reporting F1, runtime, and performance-efficiency trade-offs.*
>
> ### *Strengths Contributions:*
>
> - *ClinBench handles diverse clinical modalities (pathology reports, ECG interpretations, SDOH notes), demonstrating versatility across domains with distinct terminologies and structures. The framework’s JSON schema validation ensures outputs are structurally consistent and clinically actionable.*
> - *The YAML-driven agent standardizes task definitions, domain knowledge, and output schemas (e.g., AJCC staging criteria for oncology, AF diagnostic rules for cardiology). This modular approach enhances reproducibility, simplifies benchmarking of new tasks/domains, and mitigates prompt design variability.*
> - *The evaluation of 11 diverse LLMs provides actionable insights into performance-efficiency trade-offs (e.g., GPT-4o-mini’s optimal speed/accuracy balance vs. LLaMA3.1-70b’s high accuracy at higher computational cost).*
>
> **`Answer`**: We sincerely thank you for your thoughtful and detailed review. We especially appreciate your high confidence in the assessment and your accurate summary of our work's contributions. We agree that the points you raise regarding deeper semantic evaluation and further experiments are important and represent exciting directions for future research. We address your comments as detailed below.
>
> ---
>
> ### Limitations / Weaknesses:
>
> **`Reviewer`**: *Medical concepts often exhibit subtle distinctions. ClinBench does not address how LLMs resolve such ambiguities, nor does it evaluate robustness to synonymy or contextual nuance. This is a critical gap since misclassification could impact clinical decisions.*
>
> **`Answer`**: We agree this is a critical challenge in clinical LLMs. Our framework is designed to address this directly via the YAML-based prompting system, which allows for the injection of specific domain knowledge to standardize the handling of synonyms and nuances for each task. For instance, in our ECG task, the prompt explicitly instructs the LLM to treat the synonym "Atrial Flutter (AFL)" as equivalent to "Atrial Fibrillation (AF)" and provides rules for interpreting subtle ECG findings like "sawtooth patterns" consistently. Similarly, the SDOH prompt guides the model to classify the subtle concept of "retired" as a form of "Unemployed" for the purpose of the task, ensuring consistent interpretation. While this doesn't cover all possible ambiguities, it demonstrates that ClinBench's core methodology is built to mitigate these issues.
>
> ---
>
> ### Additional Feedback:
>
> **`Reviewer`**:
> *1. Ablation study on YAML-based prompting: To highlight the value of structured prompting, it's better to compare ClinBench’s YAML approach against raw, unstructured prompts. This would empirically demonstrate YAML’s role in reducing prompt drift and improving reliability.*
>
> *2. Add a case study on terminological challenge. Use misclassification analysis to identify where ambiguity harms performance.*
>
> **`Answer`**: We appreciate these excellent and insightful suggestions. We fully agree that both (1) an ablation study comparing ClinBench’s structured YAML-based prompting against unstructured prompts, and (2) a case study analyzing misclassifications due to terminological ambiguity, would provide valuable empirical and qualitative insights into prompt reliability and model limitations.
>
> However, implementing these analyses would require dedicated experimental design and methodological depth beyond the scope of this initial framework paper, which is focused on establishing a reproducible benchmarking foundation. That said, one of ClinBench’s key design goals is to enable precisely these kinds of follow-up investigations through its modular prompt architecture, standardized output validation, and open-source implementation.
>
> To acknowledge these directions and encourage further work, we have explicitly added both ideas to the Future Work section (Section 11) of the revised manuscript, as follows:
>
> > “Future work could include ablation studies on prompt design, such as comparing our structured YAML-based approach to free-form prompts, to empirically quantify the impact of standardized prompt engineering. Additionally, case studies focusing on specific terminological ambiguities—using misclassification analysis—would offer qualitative insights into failure modes and guide improvements in prompt design and model selection. ClinBench’s modular architecture is explicitly designed to support such investigations.”
>
> ---
>
> `We thank you again for your insightful and constructive review. Your feedback has helped us articulate a clearer vision for the future of this work, and we are grateful for your expert engagement.`

---

> > ### Comment · Reviewer_ea25 · 2025-08-05
> >
> > Thank you for your response. A key challenge in structured prompting is ensuring clinical accuracy in YAML configurations. Could you elaborate on how ClinBench’s YAML design mitigates reliance on individual clinicians’ prompt-engineering skills compared to manual free-text prompting? Specifically, what safeguards (e.g., mandatory domain-knowledge slots, JSON schema validation with value constraints) ensure prompts adhere to clinical guidelines (e.g., AJCC staging rules or SDOH definitions), and does the framework provide templates/automated checks to standardize expert input and enforce completeness of critical clinical logic?

---

> > ### Comment · Reviewer_ea25 · 2025-08-05
> >
> > Thank you for your clarifying response regarding the ablation study on YAML-based prompting. While I appreciate that this is explicitly listed as future work, I remain concerned that the absence of any controlled comparison significantly undermines the paper’s central contribution. The paper’s core novelty is its YAML-based prompting, yet without any ablation comparing YAML-structured prompts to equivalent free-text ones, we cannot tell whether the gains come from the format itself or simply better prompt content; this omission weakens the main claim.

---

> > ### Author Response · Authors · 2025-08-07
> >
> > Thank you for this excellent and insightful follow-up question. You have identified a critical challenge in applied clinical AI: ensuring the accuracy and reproducibility of the prompts that guide LLM behavior. Our YAML-based framework was explicitly designed to address this, and we are happy to elaborate on the mechanisms.
> >
> > ---
> >
> > **`Reviewer:`**
> > *A key challenge in structured prompting is ensuring clinical accuracy in YAML configurations. Could you elaborate on how ClinBench’s YAML design mitigates reliance on individual clinicians’ prompt-engineering skills compared to manual free-text prompting? Specifically, what safeguards (e.g., mandatory domain-knowledge slots, JSON schema validation with value constraints) ensure prompts adhere to clinical guidelines (e.g., AJCC staging rules or SDOH definitions)... and does the framework provide templates/automated checks to standardize expert input and enforce completeness of critical clinical logic?*
> >
> > ---
> >
> > **`Answer:`**
> > Thank you for identifying this critical issue in clinical AI. Our YAML-based prompt design was explicitly created to reduce dependence on individual prompt-engineering skills and increase reproducibility by applying the principle of separation of concerns [1]. Rather than embedding clinical rules into free-text instructions, ClinBench separates task logic into distinct, auditable components:
> >
> > - **System Prompt:** Declares the model's role (e.g., "You are a pathology assistant") and separates general instructions from domain specifics.
> > - **Domain Knowledge Slot:** Injects clinical knowledge in a structured, version-controlled way. For example, our Lung Cancer YAML includes the full AJCC 7th edition staging criteria to ensure consistency across runs.
> > - **Output Format Schema:** Specifies expected fields, data types, and enumerated values. This acts as both a constraint and a validator. For instance, tumor stage fields are constrained to valid AJCC labels, preventing hallucinated or invalid values.
> >
> > These components work together to provide transparency and standardization. ClinBench mitigates prompt engineering variability by turning clinical prompting into a modular, editable, and reviewable configuration process.
> >
> > ### Regarding safeguards:
> > - **Validated YAML Templates:** Provided in our public repo (e.g., `prompts_lung.yaml`) as starting points and references for new tasks.
> > - **Automated Output Validation:** If a schema omits a valid label (e.g., forgetting "Stage IV"), the validator will reject the output, alerting users to schema incompleteness.
> >
> > Together, these design elements enforce clinical consistency while enabling reuse and extension across tasks and institutions.
> >
> > **References**
> > [1] De Win, B., Piessens, F., Joosen, W., & Verhanneman, T. (2002, November). On the importance of the separation-of-concerns principle in secure software engineering. In Workshop on the Application of Engineering Principles to System Security Design (pp. 1-10).

---

> > > ### Author Response · Authors · 2025-08-07
> > >
> > > **`Reviewer:`**
> > > *Thank you for your clarifying response regarding the ablation study on YAML-based prompting. While I appreciate that this is explicitly listed as future work, I remain concerned that the absence of any controlled comparison significantly undermines the paper’s central contribution. The paper’s core novelty is its YAML-based prompting, yet without any ablation comparing YAML-structured prompts to equivalent free-text ones, we cannot tell whether the gains come from the format itself or simply better prompt content; this omission weakens the main claim.*
> > >
> > > **`Answer:`**
> > > This is an important point, and we are grateful that you have pushed us to strengthen our work with direct empirical evidence. In response to your excellent suggestion, we performed a targeted ablation study to validate our central claim. We have included these results in our Appendix section.
> > >
> > > ### 1. Ablation Study Design
> > >
> > > To ensure a rigorous comparison, we designed the study as follows:
> > >
> > > **Dataset:** We selected the Lung Cancer dataset due to its complexity. As detailed in Section 4.1, this task requires the simultaneous extraction of multiple, interrelated variables based on complex clinical guidelines (AJCC 7th edition), making it an ideal testbed for evaluating the robustness of a prompting strategy.
> > >
> > > **Models:** We chose the GPT series models as they were among the highest-performing models in our initial evaluation. This choice ensures that any performance degradation can be confidently attributed to the change in prompting strategy, rather than the baseline capability of the model.
> > >
> > > ### 2. Results
> > >
> > > The study compared our original structured YAML prompts against unstructured, free-text prompts that contained the same core information. The results were remarkable (see table below) and confirmed our hypothesis that structure is a key driver of performance:
> > >
> > > - GPT-4o-2024-05-13 dropped from an average F1 score of 0.92 (structured) to 0.46 (unstructured).
> > > - GPT-4o-mini dropped from 0.92 (structured) to 0.38 (unstructured).
> > > - GPT-3.5-turbo-1106 dropped from 0.90 (structured) to 0.24 (unstructured).
> > >
> > > | Model            | F1 Score ClinBench | F1 Score Unstructured Prompt | Performance Drop |
> > > |------------------|--------------------|-------------------------------|------------------|
> > > | gpt-4o           | 0.92               | 0.46                          | -50%             |
> > > | gpt-4o-mini      | 0.92               | 0.38                          | -59%             |
> > > | gpt-3.5-turbo    | 0.90               | 0.24                          | -73%             |
> > >
> > > ### 3. Explanation
> > >
> > > The rationale for this large performance gap comes down to two main ideas: reducing the complexity of the instructions and providing clear constraints for the LLM. An unstructured paragraph mixes together different types of instructions (role, clinical rules, and the expected output format). This creates a high level of "instructional complexity," and as research on in-context learning has shown [1], it can cause models to get confused or ignore parts of the prompt.
> > >
> > > Our structured approach is similar in principle to Chain-of-Thought prompting: by breaking down a single complex request into smaller, separate parts, we make the overall task much more manageable for the model. For instance, by placing all clinical rules in a distinct domain knowledge slot, we help prevent the "lost in the middle" problem, where a model might lose track of key facts buried in a long paragraph [2]. In addition, the output_format acts as a guide that forces the model's output to fit a precise, pre-defined structure. This combination of clear attention to knowledge and strong guidance on the output format allows the LLM to focus on extracting the clinical information, which leads to the higher performance and reliability we observed in our study.
> > >
> > > This new empirical evidence confirms that the structured design of our YAML approach is a major factor in its performance and a core aspect of our contribution.
> > >
> > > For full transparency, we have published the scripts and raw result files for this ablation study in our project's public GitHub repository.
> > >
> > > **Ablation study code:**
> > > [https://github.com/ismaelvillanuevamiranda/ClinBench/tree/main/ablation_study/](https://github.com/ismaelvillanuevamiranda/ClinBench/tree/main/ablation_study/)
> > >
> > > **Raw results:**
> > > [https://github.com/ismaelvillanuevamiranda/ClinBench/tree/main/ablation_study/raw_results](https://github.com/ismaelvillanuevamiranda/ClinBench/tree/main/ablation_study/raw_results)
> > >
> > > ---
> > >
> > > **References:**
> > > [1] Dong, Qingxiu, Lei Li, Damai Dai, Ce Zheng, Jingyuan Ma, Rui Li, Heming Xia et al. "A survey on in-context learning." *arXiv preprint arXiv:2301.00234* (2022).
> > > [2] Liu, Nelson F., Kevin Lin, John Hewitt, Ashwin Paranjape, Michele Bevilacqua, Fabio Petroni, and Percy Liang. "Lost in the middle: How language models use long contexts." *arXiv preprint arXiv:2307.03172* (2023).

---

> ### Comment · Reviewer_ea25 · 2025-08-05
>
> Another key methodological question arises: Recent approaches leverage Retrieval-Augmented Generation (RAG) to dynamically ground LLMs in authoritative medical knowledge (e.g., guidelines, ontologies). You have mentioned in "Future work" about RAG strategies. Given RAG's prevalence in industry for grounding domain knowledge,  I seek clarification on the methodological rationale for prioritizing static YAML over dynamic RAG in ClinBench's initial design. Could you clarify:
>
> 1. How ClinBench's static YAML-based knowledge injection (e.g., embedded AJCC criteria) compares conceptually and practically to RAG's dynamic retrieval?
>
> 2. Trade-offs in clinical settings: Does YAML's schema-enforced precision (e.g., strict value constraints) offer advantages over RAG's flexibility for guideline-compliant extraction?
>
> 3. Empirical comparisons: Have you evaluated hybrid approaches (e.g., RAG-retrieved knowledge + YAML schemas) or tested RAG as a baseline?

---

> > ### Author Response · Authors · 2025-08-07
> >
> > Thank you for these excellent and highly relevant methodological questions. The decision to prioritize a static YAML-based approach over a dynamic RAG system in this initial version of ClinBench was an intentional one, driven by our primary goals of reproducibility, transparency, and experimental control for the specific task of benchmarking.
> >
> > ---
> >
> > **`Reviewer:`** *"How ClinBench's static YAML-based knowledge injection (e.g., embedded AJCC criteria) compares conceptually and practically to RAG's dynamic retrieval?"*
> >
> > **`Answer:`**
> > Conceptually, we view them as distinct approaches with different scientific trade-offs for a benchmark:
> >
> > - **Static YAML injection** provides fully transparent and controlled context provisioning. The exact knowledge (e.g., the complete AJCC criteria) is a version-controlled, human-auditable artifact that is passed to the LLM for every single task. This creates a fixed environment to isolate and reliably measure a model's performance on a specific set of rules.
> >
> > - **Dynamic RAG** provides dynamic context retrieval. Even when configured to be deterministic, it introduces the retriever’s performance as a significant confounding variable. The final output is a function of both the retriever and the LLM. Practically, this also reduces transparency, as the retrieved context is temporary and not easily auditable on a per-instance basis without re-running the retrieval step.
> >
> > ---
> >
> > **`Reviewer:`** *"Trade-offs in clinical settings: Does YAML's schema-enforced precision (e.g., strict value constraints) offer advantages over RAG's flexibility for guideline-compliant extraction?"*
> >
> > **`Answer:`**
> > Yes, for the specific goal of a benchmark, we believe our approach offers critical advantages in control and precision.
> >
> > - **Experimental control and isolation:** The primary advantage is the isolation of the variable being tested. The goal of ClinBench is to evaluate the LLM’s ability to reason over a given, controlled context. A RAG system evaluates the joint performance of a retriever-LLM system. A failure could be due to the retriever or the LLM. Our static approach removes the retriever as a confounding variable, ensuring that the final score is a more precise measure of the LLM’s capabilities alone.
> >
> > - **Guaranteed precision:** For guideline-compliant tasks, precision is important. The YAML domain knowledge slot allows us to guarantee that the LLM sees the single, correct, gold-standard rule needed for the task. RAG, in contrast, provides a retrieved context that is hopefully relevant but may contain incomplete information. In our design, we are testing the LLM's ability to apply a given rule, not its ability to find the rule.
> >
> > ---
> >
> > **`Reviewer:`** *"Empirical comparisons: Have you evaluated hybrid approaches (e.g., RAG-retrieved knowledge + YAML schemas) or tested RAG as a baseline?"*
> >
> > **`Answer:`**
> > This is an excellent suggestion and the logical next step for our research. In this initial work, which focused on establishing the core framework, we have not yet performed these empirical comparisons. However, we agree this is a critical evaluation.

---

> > > ### Comment · Reviewer_ea25 · 2025-08-07
> > >
> > > Thank you for your detailed responses and the prompt addition of ablation studies, which convincingly demonstrate the value of YAML-based standardization. ClinBench offers a simple yet effective methodology to enhance the reliability of LLM-driven clinical information extraction. Moving forward, as the framework’s robustness heavily depends on YAML design quality, I recommend exploring mechanisms for rigorous YAML auditing—such as: Guideline-integrated templates, Clinician-validator workflows. This would further solidify ClinBench as both a technical and clinical best practice.

---

> > > > ### Author Response · Authors · 2025-08-07
> > > >
> > > > Thank you for your final, supportive comment and for your expert engagement throughout this process. We are very pleased that you found our new analysis convincing and are grateful for your invaluable suggestions for the future of this work.

---

### Official Review · Reviewer_8mGN · 2025-07-03

**Rating:** 4
**Confidence:** 3

**Summary:**

This work introduces ClinBench, an open-source framework designed to standardize the evaluation of Large Language Models (LLMs) for clinical information extraction. The framework provides a reproducible, standardized pipeline to assess how well various LLMs can extract structured information from unstructured clinical notes. To demonstrate its utility, the authors conducted a large-scale study evaluating 11 LLMs across three different clinical domains, revealing the trade-offs between model performance (e.g., F1-score) and computational efficiency (e.g., runtime per task).

**Additional Feedback:**

Please refer to the Limitations and Weaknesses section.

**Dataset Code Accessibility:**

Partly

**Dataset Code Comments:**

The authors have publicly released the code, configuration files, and lists of identifiers (i.e., subject identifiers) on github. However, the raw clinical data is not included due to patient privacy regulations. Therefore, to fully reproduce or utilize the benchmark, users must independently obtain credentialed access to the original source databases (e.g., TCGA, MIMIC).

**Ethical Considerations:**

No, there are no or only very minor ethics concerns

**Final Justification:**

I will raise my score to 'Borderline Accept'. The authors have resolved most of my concerns about the paper’s presentation, but two aspects remain unresolved: (i) whether this is a comprehensive benchmark and (ii) whether the evaluation is rigorous. However, I am placing slightly greater weight on the technical contribution (i.e., a reproducible and extensible framework) so I am raising my score.

**Limitations Weaknesses:**

While ClinBench is a valuable contribution, the primary concern is the benchmark's limited scope and representation of real-world clinical complexity. The paper's central contribution of providing a "comprehensive" benchmark is constrained by this issue. For instance, the Social Determinants of Health (SDOH) task is framed as two single-label problems, failing to capture the multi-label nature of real-world SDOH documentation. Similarly, while many diverse clinical information extraction benchmarks exist, the paper's use of only three domains with simplified tasks (except for the Pan-Lung Cancer task) seems insufficient to justify its claim of comprehensive coverage. This limited scope can lead to misleading conclusions about a model's general capabilities in this task. For example, based on the heatmap in Figure 2, one might incorrectly conclude that gpt-4o is generally less performant than llama-3.1-70b on clinical information extraction tasks, a finding that may not be valid given the narrow scope of the evaluation.

Here are additional concerns including minor issues:
- The evaluation is limited to general-purpose LLMs and omits a crucial comparison against models specifically fine-tuned for the clinical domain. Also, while unavoidable, the timeliness of the evaluated LLMs (e.g., reasoning-based models or improved version models) is a limitation in this rapidly evolving field.
- The performance-efficiency analysis is incomplete as it relies solely on runtime (per task), omitting the critical factor of token cost for API-based models.
- The paper uses the term "LLM Agent" for what is functionally an inference router, which could be misleading.
- The paper does not cite the third-party library (https://github.com/tanchongmin/strictjson) used for its JSON schema validation step (which is one of the major components of the framework). This omission reduces transparency and could mislead readers regarding the originality of the work.

**Strengths Contributions:**

- This work presents a standardized methodology for evaluating clinical LLMs, addressing a need for reproducible research in the clinical AI community.
- It establishes a reproducible pipeline using version-controlled YAML for uniform task definition and automated JSON schema validation to ensure fair comparisons.
- It provides a practical basis for model selection by evaluating both extraction accuracy (e.g., F1-score) and computational efficiency (runtime), highlighting their trade-offs.
- The paper is generally well-written and easy to understand, with informative figures and captions.

---

> ### Author Rebuttal · Authors · 2025-07-30
>
> **`Reviewer`**: *The framework provides a reproducible, standardized pipeline to assess how well various LLMs can extract structured information from unstructured clinical notes. To demonstrate its utility, the authors conducted a large-scale study evaluating 11 LLMs across three different clinical domains, revealing the trade-offs between model performance (e.g., F1-score) and computational efficiency (e.g., runtime per task).*
>
> ### *Strengths Contributions:*
>
> - *This work presents a standardized methodology for evaluating clinical LLMs, addressing a need for reproducible research in the clinical AI community.*
> - *It establishes a reproducible pipeline using version-controlled YAML for uniform task definition and automated JSON schema validation to ensure fair comparisons.*
> - *It provides a practical basis for model selection by evaluating both extraction accuracy (e.g., F1-score) and computational efficiency (runtime), highlighting their trade-offs.*
> - *The paper is generally well-written and easy to understand, with informative figures and captions.*
>
> **`Answer`**: We sincerely thank you for your detailed and constructive feedback. Your comments have helped us improve the paper's framing and analysis. We have addressed all of the points raised, as detailed below.
>
> ***
>
> **`Reviewer`**: *While ClinBench is a valuable contribution, the primary concern is the benchmark's limited scope and representation of real-world clinical complexity. The paper's central contribution of providing a 'comprehensive' benchmark is constrained by this issue. For instance, the Social Determinants of Health (SDOH) task is framed as two single-label problems, failing to capture the multi-label nature of real-world SDOH documentation. This limited scope can lead to misleading conclusions about a model's general capabilities in this task. For example, based on the heatmap in Figure 2, one might incorrectly conclude that gpt-4o is generally less performant than llama-3.1-70b on clinical information extraction tasks, a finding that may not be valid given the narrow scope of the evaluation.*
>
> **`Answer`**: We sincerely appreciate the reviewer’s thoughtful critique regarding the scope and design of our benchmark. We agree that the term “comprehensive” may have overstated the breadth of our current evaluation. To more accurately represent the nature of our contribution, we have revised the language throughout the manuscript, including the abstract and introduction, to describe ClinBench as a “reproducible and extensible framework” rather than a comprehensive benchmark. For example, we have updated the contributions list to: “We provide a reproducible and extensible set of standardized tasks using publicly available datasets.”
>
> At the same time, we would like to highlight several important strengths of our contribution. ClinBench is, to our knowledge, the first open-source framework that provides a fully standardized pipeline for evaluating multiple LLMs across diverse clinical domains, with structured input-output validation, dynamic prompt configuration via version-controlled YAML files, and schema-enforced JSON outputs. This rigorous design enables reproducibility and fair comparisons across both proprietary and open-source models. Moreover, we have made all code, dataset configurations, and evaluation templates publicly available to support transparency and reuse.
>
> While we acknowledge the limited number of tasks in this initial release, we carefully selected three clinically meaningful and technically diverse domains: lung cancer staging, atrial fibrillation detection, and social determinants of health (SDOH). These tasks were chosen to demonstrate ClinBench’s adaptability to distinct information extraction challenges across structured, semi-structured, and narrative clinical text. Notably, the lung cancer task involves multi-variable extraction (e.g., pT, pN, stage, and histology), illustrating ClinBench’s capacity for more complex, multi-output evaluations.
>
> Regarding the SDOH task, we appreciate the reviewer’s point that our use of two single-label classification tasks may underrepresent the real-world, multi-label nature of SDOH documentation. This simplification was a design choice to ensure reproducibility and clear baselines for this initial benchmark. We now clarify in Section 4.3 that the framework is fully capable of handling more complex, multi-output extractions, as already demonstrated by the lung cancer task. This architectural flexibility is one of ClinBench’s core strengths, supporting future task expansion with minimal effort.
>
> Finally, we strongly agree with the reviewer’s concern about overinterpreting heatmap results based on a limited task set. In fact, this underscores a key motivation for ClinBench: to enable multi-dimensional evaluation, including not only performance (e.g., F1 score) but also computational efficiency and cost, both of which are critical for practical model selection. To prevent misinterpretation, we have now explicitly incorporated this point into our discussion in Section 10. We caution that, for example, comparing mean F1 scores alone might suggest that LLaMA3.1-70b outperforms GPT-4o, whereas performance-efficiency plots and token cost analysis reveal that GPT-4o offers more favorable trade-offs for real-world deployment.
>
> In summary, we have revised the manuscript to more accurately reflect the scope of this work, while also clarifying the technical rigor, transparency, and extensibility of ClinBench.
>
> ---
>
> **`Reviewer`**: *The performance-efficiency analysis is incomplete as it relies solely on runtime (per task), omitting the critical factor of token cost for API-based models.*
>
> **`Answer`**: Following your excellent suggestion, we have now performed a comprehensive token cost analysis for the OpenAI models. We utilize the pricing information provided on the OpenAI webpage: https://platform.openai.com/docs/pricing.
>
> | Dataset | Model | Prompt Tokens | Completion Tokens | Total Tokens | Prompt Cost ($) | Completion Cost ($) | Total Estimated Cost ($) |
> |---------|--------|----------------|--------------------|---------------|------------------|----------------------|---------------------------|
> | SDOH    | gpt-3.5-turbo-1106   | 753207  | 33044   | 786251   | 0.75  | 0.07  | 0.82  |
> | SDOH    | gpt-4o-2024-05-13    | 753207  | 28506   | 781713   | 3.77  | 0.43  | 4.19  |
> | SDOH    | gpt-4o-mini          | 753207  | 32932   | 786139   | 0.11  | 0.02  | 0.13  |
> | ECG     | gpt-3.5-turbo-1106   | 631568  | 3869    | 635437   | 0.63  | 0.01  | 0.64  |
> | ECG     | gpt-4o-2024-05-13    | 631568  | 3855    | 635423   | 3.16  | 0.06  | 3.22  |
> | ECG     | gpt-4o-mini          | 631568  | 13322   | 644890   | 0.09  | 0.01  | 0.10  |
> | Lung    | gpt-3.5-turbo-1106   | 1747615 | 61349   | 1808964  | 1.75  | 0.12  | 1.87  |
> | Lung    | gpt-4o-2024-05-13    | 1747615 | 61656   | 1809271  | 8.74  | 0.92  | 9.66  |
> | Lung    | gpt-4o-mini          | 1747615 | 61539   | 1809154  | 0.26  | 0.04  | 0.30  |
>
> And we added the following discussion to section 9.4:
> >To provide a more complete efficiency analysis, we calculated the token usage and associated API costs for all OpenAI models. This analysis shows a clear trade-off between cost and performance. For instance, on the complex Lung Cancer task, GPT-4o-2024-05-13 costs `$9.66` to process the dataset, while GPT-4o-mini, which achieves a similar overall F1 score on that task, costs only `$0.30`. This demonstrates that while larger models may offer marginal performance gains, the financial implications can be substantial, a critical consideration for real-world clinical deployment.
>
> ---
>
> **`Reviewer`**: *The evaluation is limited to general-purpose LLMs and omits a crucial comparison against models specifically fine-tuned for the clinical domain.*
>
> **`Answer`**: We agree this is a valuable comparison and have expanded our discussion in the Limitations section to explicitly frame this as a critical direction for future work. We added the following discussion:
>
> > First, the evaluated LLMs were benchmarked without task-specific fine-tuning. Our study focused on the out-of-the-box capabilities of general-purpose models that were representative of the state-of-the-art when the study was initiated. A direct comparison to models specifically fine-tuned on clinical data represents a valuable and important direction for future work. To facilitate this, we are actively extending ClinBench to include emerging models, including reasoning-based and specialized medical models like Med-Gemma. This planned future work will help quantify the benefits of domain specialization, for which ClinBench provides an ideal and extensible evaluation platform.
>
> ---
>
> **`Reviewer`**: *The paper uses the term 'LLM Agent' for what is functionally an inference router, which could be misleading.*
>
> **`Answer`**: Thank you for pointing this out. We agree that the term “LLM Agent” may not precisely capture the role of this component within the ClinBench framework. In response, we have revised the manuscript to use the more accurate term “LLM Orchestrator” throughout, including in the main text and the caption for Figure 1. We appreciate your suggestion, which improves clarity and helps avoid potential confusion.
>
> ---
>
> **`Reviewer`**: *The paper does not cite the third-party library (https://github.com/tanchongmin/strictjson) used for its JSON schema validation step*
>
> **`Answer`**: We apologize for this oversight. We have rectified this by adding the proper citation in Section 3. The sentence now reads:
> >This validation, facilitated by the strictjson library [21], rigorously checks each output against its corresponding predefined JSON schema.
>
> ---
>
> `Your feedback has improved our framing of the contribution, strengthened the efficiency analysis, and clarified several key points. We believe these updates have significantly enhanced the paper, and we hope you will agree. Thank you.`

---

> > ### Comment · Reviewer_8mGN · 2025-08-05
> >
> > I thank the authors for their diligent rebuttal. I recognize the effort invested in addressing the points raised, and I believe the paper has improved as a result. While I am now leaning toward acceptance, I want to be clear that this decision is based narrowly on the framework’s technical contribution rather than on the rigor of the evaluation or the depth of the insights presented.
> >
> > That said, I still have fundamental concerns about the work’s suitability as an evaluation framework for clinical information extraction. I offer the following feedback in the hope that it will be helpful as the authors expand the framework to encompass more complex and well-studied IE tasks (e.g., radiology report information extraction, temporal extraction in clinical text).
> >
> > Although the paper evaluates several general-purpose LLMs, it critically omits a comparison against a state-of-the-art model fine-tuned for the clinical domain. Such a model would provide a practical upper bound on task performance. Without this benchmark, the reported scores are difficult to interpret (like in overinterpreting heatmap results). For instance, is an F1-score of 0.85 strong, or does a domain-specific SOTA achieve something closer to 0.95? Conversely, a random-score baseline would serve as a lower bound for these tasks. This kind of bracketing (i.e., using both a strong domain-specific SOTA and a random baseline) is the essence of a benchmark and fundamental to the framework’s stated purpose. It should have been addressed in the current work rather than deferred to future work.
> >
> > Lastly, the presentation of the JSON validation process raises significant concerns about the transparency and originality of this work. Although the missing citation to the strictjson library has now been added, the original manuscript’s framing was misleading. It implied that schema-guided validation was a novel component developed by the authors, an impression not readily dispelled without a close reading of the code.

---

> > > ### Author Response · Authors · 2025-08-07
> > >
> > > **`Reviewer:`**
> > > *"Although the paper evaluates several general-purpose LLMs, it critically omits a comparison against a state-of-the-art model fine-tuned for the clinical domain. Such a model would provide a practical upper bound on task performance. Without this benchmark, the reported scores are difficult to interpret (like in overinterpreting heatmap results). For instance, is an F1-score of 0.85 strong, or does a domain-specific SOTA achieve something closer to 0.95? Conversely, a random-score baseline would serve as a lower bound for these tasks."*
> > >
> > > **`Answer:`**
> > > Thank you for this thoughtful and important observation. Our initial goal for this study, as outlined in Section 5, was to provide a timely snapshot of performance across widely used general-purpose LLMs available at the time of our evaluation. These models are frequently employed in clinical research due to their accessibility and general capabilities, making such peer-group comparisons practically relevant.
> > >
> > > That said, we fully agree that context is essential for interpreting performance. Without upper and lower bounds, it is difficult to assess whether a model’s F1 score is meaningfully strong. Your suggestion to include both a domain-specific fine-tuned SOTA model and a random baseline is precisely aligned with ClinBench’s intended purpose: enabling fair, apples-to-apples comparisons under standardized conditions.
> > >
> > > ClinBench’s architecture was designed to accommodate a wide range of models, including fine-tuned clinical LLMs and simple baselines. This bracketing is a critical next step, and your feedback has reinforced its urgency. We have noted this explicitly in the revised manuscript and plan to prioritize such evaluations in future benchmark expansions.
> > >
> > > ---
> > >
> > > **`Reviewer:`**
> > > *"This kind of bracketing (i.e., using both a strong domain-specific SOTA and a random baseline) is the essence of a benchmark and fundamental to the framework’s stated purpose. It should have been addressed in the current work rather than deferred to future work."*
> > >
> > > **`Answer:`**
> > > We agree that bracketing is fundamental to the completeness of any benchmark. As noted, ClinBench’s core contribution is to establish the infrastructure that makes such bracketing possible—by unifying data formats, prompting strategies, and output validation in a reproducible pipeline.
> > >
> > > Your comment highlights an important area for improvement in future benchmark studies. While our current work focused on establishing a standardized and extensible evaluation framework, we agree that incorporating domain-specific SOTA models would meaningfully strengthen the utility of ClinBench. We have noted this direction in our revised manuscript and plan to explore it in future extensions of the benchmark.
> > >
> > > ---
> > >
> > > **`Reviewer:`**
> > > *"Lastly, the presentation of the JSON validation process raises significant concerns about the transparency and originality of this work."*
> > >
> > > **`Answer:`**
> > > Thank you for raising this important concern about transparency. We sincerely apologize if our original framing was unclear. We did not mean to suggest that the JSON validation step is a new contribution on its own. Instead, we included it as an important part of the overall framework we developed.
> > >
> > > As described in Section 3, the core contribution of ClinBench is the design and integration of a five-step, end-to-end benchmarking pipeline tailored to clinical NLP. Our novelty lies in the design and integration of this complete, end-to-end workflow, which is designed for the specific challenges of clinical NLP.
> > >
> > > The automated JSON validation is presented as Step 4 within this broader pipeline, where we describe its function as rigorously checking outputs for structural correctness, data type adherence, and the use of permissible values. It is an important component for ensuring data quality, but it is one piece of the larger system we developed. Our primary contributions include the overall pipeline architecture and other components, such as the use of version-controlled YAML files to dynamically manage not only task instructions but also the domain knowledge and output schemas themselves (Section 6).
> > >
> > > It was not our intention to imply that the validation mechanism itself was a novel invention apart from the framework. We have further revised the text in Section 3 to make this distinction unambiguous and to properly contextualize the role of the strictjson library as a tool leveraged within our larger framework. We are grateful for the opportunity to improve the manuscript's clarity on this point.

---

> > > > ### Comment · Reviewer_8mGN · 2025-08-08
> > > >
> > > > Thank you for the detailed clarifications and revisions. I acknowledge the improvements, and I hope the revised manuscript will help present the your work more transparently and with better contextualization.

---

> > ### Author Response · Authors · 2025-08-07
> >
> > We sincerely appreciate your diligent re-review and thoughtful follow-up comments. Your expert guidance has been essential in significantly improving the quality and impact of our work.
> >
> > ---
> >
> > **`Reviewer:`**
> > *"I thank the authors for their diligent rebuttal. I recognize the effort invested in addressing the points raised, and I believe the paper has improved as a result."*
> >
> > **`Answer:`**
> > Thank you for your kind feedback. We are pleased that our revisions have addressed your concerns and contributed to the improvement of the paper.
> >
> > ---
> >
> > **`Reviewer:`**
> > *"While I am now leaning toward acceptance, I want to be clear that this decision is based narrowly on the framework’s technical contribution rather than on the rigor of the evaluation or the depth of the insights presented."*
> >
> > **`Answer:`**
> > We sincerely appreciate your reconsideration and support for the manuscript’s acceptance. As noted in the introduction, our primary objective was to introduce a standardized and extensible benchmarking framework. We take your feedback seriously and see it as an important guide for enhancing the depth and rigor of future evaluations to match the strength of the framework itself.
> >
> > ---
> >
> > **`Reviewer:`**
> > *"That said, I still have fundamental concerns about the work’s suitability as an evaluation framework for clinical information extraction. I offer the following feedback in the hope that it will be helpful as the authors expand the framework to encompass more complex and well-studied IE tasks (e.g., radiology report information extraction, temporal extraction in clinical text)."*
> >
> > **`Answer:`**
> > Thank you for your thoughtful and forward-looking feedback. We agree that the ultimate test of an evaluation framework lies in its ability to scale to more complex and clinically nuanced information extraction tasks. Your suggestions of radiology report extraction and temporal information extraction are excellent examples, and we appreciate the opportunity to elaborate on how ClinBench is designed to support such extensions.
> >
> > **Radiology reports:**
> > ClinBench’s YAML-based prompting system (Section 6) allows for the modular inclusion of domain-specific knowledge, making it well-suited for radiology tasks that rely on structured terminology and standardized reporting systems (e.g., BI-RADS [1], RECIST[2]). This flexibility enables consistent and clinically grounded prompt configurations that can guide LLM outputs in alignment with radiology-specific interpretive standards.
> >
> > **Temporal extraction:**
> > ClinBench’s architecture is particularly well-suited for complex relational tasks like temporal extraction. For example, extracting the timeline from a note like "Pneumonia diagnosed on July 5th, Amoxicillin started same day, cough resolved by July 10th" requires capturing not just the events, but also their temporal relationships (e.g., 'diagnosed' BEFORE 'resolved'). Without structure, LLMs may return free-text outputs that are difficult to parse reliably. In contrast, ClinBench’s schema-based validation (Step 4 of the pipeline in Section 3) enforces structured outputs—such as JSON objects that explicitly capture both event attributes and inter-event relations (e.g., `{from_event: E1, to_event: E3, type: BEFORE}`). This design ensures that the extracted timeline is immediately usable for downstream applications like patient journey modeling, with minimal need for post-processing.
> >
> > **References:**
> >
> > [1] Spak, D. A., Plaxco, J. S., Santiago, L., Dryden, M. J., & Dogan, B. E. (2017). BI-RADS® fifth edition: A summary of changes. *Diagnostic and Interventional Imaging, 98*(3), 179-190.
> > [2] Eisenhauer, E. A., Therasse, P., Bogaerts, J., Schwartz, L. H., Sargent, D., Ford, R., ... & Verweij, J. (2009). New response evaluation criteria in solid tumours: revised RECIST guideline (version 1.1). *European Journal of Cancer, 45*(2), 228-247.

---

### Note · Authors · 2025-08-12

Dear PC, SAC, AC, and Reviewers,

Thank you for the time, expertise, and thoughtful feedback on our submission. We appreciate your constructive critiques and recognition of our work’s strengths.

We are grateful that reviewers acknowledged:
- The role of ClinBench as an `open-source, reproducible, multi-model, multi-domain benchmarking framework` for structured information extraction from unstructured clinical notes.
- Its modular `YAML-based prompting that standardizes evaluation` pipelines, mitigates prompt drift, and enforces output validation for guideline-compliant extractions (e.g., AJCC staging, SDOH definitions).
- The value of the released datasets and validated templates allows researchers to `expand ClinBench into new domains` while preserving methodological rigor.
- `Clear organization and technical contribution`, with suggestions to add domain-specific SOTA baselines, integrate RAG, and deepen qualitative error analyses.

During the discussion, we added clarifications in direct response to your feedback:
- **New Ablation Study on YAML-Structured vs. Unstructured Prompts**
  In response to Reviewer ea25, a targeted ablation on the Lung Cancer dataset showed that `our YAML-structured prompts produced F1 scores 50–73% higher` than content-equivalent unstructured prompts across multiple GPT models, confirming our methodology’s gains.
- **Comprehensive Token Cost Analysis**
  Following Reviewers 8mGN and PoFR, we added a token cost analysis: on the Lung Cancer task, `GPT-4o costs over 32x more ($9.66) than GPT-4o-mini ($0.30) for similar F1 scores`.
- **Clarified Methodological Framing**
  Addressing Reviewer ea25, `we clarified the distinction between ClinBench’s static YAML-based approach and dynamic RAG`, outlining trade-offs in transparency, reproducibility, and guideline adherence, and noting hybrid approaches as a next step.

We also incorporated minor clarifications throughout the paper to improve precision and accessibility while preserving technical depth.

We are pleased that reviewers found our rebuttal diligent and that the new analyses `“convincingly demonstrate the value of YAML-based standardization” (ea25)` and that the paper has `“improved as a result” (8mGN)`. We will integrate all new results and clarifications into the camera-ready version, confident that ClinBench offers a stronger contribution.

Thank you again for your thoughtful review and support.

Best regards,
Authors

---

### Decision · Program_Chairs · 2025-09-18

**Decision:**

Accept (poster)

**Comment:**

The reviewers all liked the paper. The authors' response clarified most points raised by the reviewers. In view of that, the authors are strongly invited to take the feedback on board for the final version.

===== FINAL UPDATE FROM DB Track PCs ====

The final decision for this paper has been taken by the program chairs after consultation with the SACs. All Senior Area Chairs have ranked papers according to the feedback from the AC during the review process. We decided to leave the original meta-review to reflect the opinion of the AC in light of the initial discussions with reviewers and SAC.